# A common molecular logic determines embryonic stem cell self-renewal and reprogramming

Sara-Jane Dunn[1,2,†] [iD], Meng Amy Li[2,†] [iD], Elena Carbognin[3] [iD], Austin Smith[2,4,*] [iD] &
Graziano Martello[3,**] [iD]

## Abstract

**During differentiation and reprogramming, new cell identities are generated by reconfiguration of gene regulatory networks. Here, we combined automated formal reasoning with experimentation to expose the logic of network activation during induction of naïve pluripotency. We find that a Boolean network architecture defined for maintenance of naïve state embryonic stem cells (ESC) also explains transcription factor behaviour and potency during resetting from primed pluripotency. Computationally identified gene activation trajectories were experimentally substantiated at single-cell resolution by RT–qPCR. Contingency of factor availability explains the counterintuitive observation that Klf2, which is dispensable for ESC maintenance, is required during resetting. We tested 124 predictions formulated by the dynamic network, finding a predictive accuracy of 77.4%. Finally, we show that this network explains and predicts experimental observations of somatic cell reprogramming. We conclude that a common deterministic program of gene regulation is sufficient to govern maintenance and induction of naïve pluripotency. The tools exemplified here could be broadly applied to delineate dynamic networks underlying cell fate transitions.**

**Keywords** abstract boolean network; formal verification; maintenance and reprogramming; naive pluripotency; transcription factor network modelling
**Subject Categories** Stem Cells; Systems & Computational Biology; Transcription
**The EMBO Journal (2019) 38: e100003**

See also: **OJL Rackham & JM Polo** (January 2019)

## Introduction

Over the last 10 years, a multitude of protocols have been developed that allow the conversion of one cell type into another (Graf &

Enver, 2009). Most of these strategies rely on the forced expression of transcription factors (TFs) highly expressed by the target cell type that have either been chosen empirically or, recently, with the aid of computational tools such as CellNet or Mogrify (Cahan *et al*, 2014; Rackham *et al*, 2016; Radley *et al*, 2017). Despite the large amount of transcriptomic data available for such conversions, our understanding of the dynamics and logic followed by cells during reprogramming and transdifferentiation remains fragmentary.

The most studied cell fate transition is the generation of murine-induced pluripotent stem cells (iPSCs) from somatic cells (Takahashi & Yamanaka, 2006). Bona fide iPSCs are, like murine embryonic stem cells (ESCs), competent to form blastocyst chimaeras and are considered to occupy a state of naïve pluripotency similar to that in the pre-implantation embryo (Nichols & Smith, 2009; Boroviak *et al*, 2015). This unique identity is determined by a self-reinforcing interaction network of TFs. Experimental and computational efforts have led to circuitry mapping of the core TF program that maintains ESC self-renewal under defined conditions (Chen *et al*, 2008; Niwa *et al*, 2009; MacArthur *et al*, 2012; Dunn *et al*, 2014; Herberg & Roeder, 2015; Rue & Martinez Arias, 2015; Yachie-Kinoshita *et al*, 2018).

We previously applied a mathematical and computational modelling approach based on automated formal reasoning to elucidate the dynamic regulatory network architecture for self-renewing mouse ESCs (Dunn *et al*, 2014; Yordanov *et al*, 2016). A minimal interaction network of 12 components was found to recapitulate a large number of observations concerning naïve state maintenance and successfully predicted non-intuitive responses to compound genetic perturbations (Dunn *et al*, 2014).

Forced expression of several components of this core TF network in various cell types leads to a state of induced pluripotency (Takahashi & Yamanaka, 2006; Nakagawa *et al*, 2007; Silva *et al*, 2008; Feng *et al*, 2009; Hanna *et al*, 2009; Han *et al*, 2010a; Buganim *et al*, 2012; Tang *et al*, 2012; O'Malley *et al*, 2013; Stuart *et al*, 2014; Sone *et al*, 2017). Accumulating evidence suggests that cells progress through defined stages, with a final transition entailing the hierarchical activation and stabilisation of the naïve

1 Microsoft Research, Cambridge, UK
2 Wellcome-MRC Cambridge Stem Cell Institute, University of Cambridge, Cambridge, UK
3 Department of Molecular Medicine, University of Padua, Padua, Italy
4 Department of Biochemistry, University of Cambridge, Cambridge, UK
*Corresponding author. Tel: +44 1223 760233; E-mail: austin.smith@cscr.cam.ac.uk
**Corresponding author. Tel: +39 049 8276088; E-mail: graziano.martello@unipd.it
†These authors contributed equally to this work

pluripotency TF network (Mikkelsen *et al*, 2008; Silva *et al*, 2009; Han *et al*, 2010b; Samavarchi-Tehrani *et al*, 2010; Buganim *et al*, 2012; Golipour *et al*, 2012; Di Stefano *et al*, 2013, 2016; O'Malley *et al*, 2013; Tanabe *et al*, 2013). However, it is not clear if cells undergoing successful conversion follow a deterministic trajectory of gene activation, defined by the naïve pluripotency TF network architecture, or if genes are activated in random sequence.

A tractable experimental system with which to investigate activation of naïve pluripotency is the resetting of post-implantation epiblast stem cells (EpiSCs; Guo *et al*, 2009). EpiSCs are related to gastrulation stage epiblast (Kojima *et al*, 2014; Tsakiridis *et al*, 2014). They represent a primed state of pluripotency, developmentally downstream of the naïve state and unable to contribute substantially to blastocyst chimaeras (Nichols & Smith, 2009). EpiSCs exhibit distinct growth factor dependency, transcriptional and epigenetic regulation compared to ESCs. They self-renew when cultured in defined media containing FGF2 and ActivinA (F/A) and lack significant expression of most functionally defined naïve pluripotency factors (Brons *et al*, 2007; Tesar *et al*, 2007; Guo *et al*, 2009). EpiSC resetting proceeds over 6–8 days, much faster than somatic cell reprogramming, and entails primarily the activation and consolidation of the naïve pluripotency identity (Hall *et al*, 2009; Festuccia *et al*, 2012; Gillich *et al*, 2012; Martello *et al*, 2013). In addition, EpiSC resetting does not require a complex reprogramming cocktail. The activation of Jak/Stat3 signalling (Han *et al*, 2010a; Yang *et al*, 2010; Bernemann *et al*, 2011a) or forced expression of a single naïve TF factor (Guo *et al*, 2009; Silva *et al*, 2009; Han *et al*, 2010a) is sufficient to mediate reprogramming in combination with dual inhibition (2i) of the Erk pathway and glycogen synthase kinase-3 (GSK3; Ying *et al*, 2008).

In this study, we undertook an iterative computational and experimental approach to test the hypothesis that a common network is sufficient to govern both naïve state maintenance and induction. Focusing on EpiSC resetting, we investigated whether naïve state induction follows an ordered sequence of network component activation. By refining our understanding of the network governing this process, we sought to delineate transcription factors crucial for the execution of EpiSC resetting, and identify synergistic combinations that accelerate resetting kinetics. Finally, we extended the approach to investigate whether the same network architecture is operative in somatic cell reprogramming.

## Results

### Deriving a set of network models consistent with EpiSC resetting

We previously studied the TF network controlling maintenance of naïve pluripotency through a combined computational and experimental approach (Dunn *et al*, 2014). Our methodology is based on the definition of relevant network components derived from functional studies in the literature, and the identification of "possible" interactions between these components (Fig 1A). Possible interactions are inferred based on gene expression correlation using the Pearson coefficient as a metric (Materials and Methods) and are used to define a set of alternative concrete Boolean network models, each with unique topology. We refer to this set of concrete models as an Abstract Boolean Network (ABN). This formalism allows us to

navigate some of the uncertainty in the interactions that may exist between network components, which can arise due to noisy or conflicting data. We then define a set of experimental results, such as the effect of genetic perturbations, which serve as constraints to identify those models from the ABN that recapitulate expected behaviour. The Reasoning Engine for Interaction Networks (RE:IN, www.research.microsoft.com/rein) is software based on automated formal reasoning, developed to synthesise only those concrete models that are provably consistent with the experimental constraints (Dunn *et al*, 2014; Yordanov *et al*, 2016). The set of consistent models is defined as a constrained Abstract Boolean Network (cABN), which is subsequently used to generate predictions of untested molecular and cellular behaviour. Our approach differs from typical modelling strategies in that we do not generate a single network model, but rather a set of models, which individually are consistent with known behaviours. We formulate predictions of untested behaviour only when all models agree, such that predictions are consistent with the limits of current understanding. This is important because different network models can recapitulate the same experimental observations, and one should not be prioritised over another. Whenever predictions are falsified by new experimental results, it is possible to refine the cABN by incorporating the new findings as additional constraints (Fig 1A). The refined cABN is then used to generate further predictions.

For the present study, we first refined the cABN describing maintenance of naïve pluripotency by adding further expression profiles generated using RNA sequencing and RT–qPCR to the five datasets used previously to infer possible interactions (Dunn *et al*, 2014) and by using an updated version of RE:IN (Yordanov *et al*, 2016; Materials and Methods). A Pearson correlation threshold of 0.832 was sufficient to define an ABN consistent with observations of maintenance of naïve pluripotency (Appendix Fig S1A–C). We identified required and disallowed interactions from this ABN to define the 0.832 cABN (Fig 1B), which we subsequently tested against new gene perturbation experiments in mouse ESCs (Appendix Fig S1D) and observed a significant increase in prediction accuracy over the previous version (Dunn *et al*, 2014). We therefore used the 0.832 cABN as the starting point for analysis of EpiSC resetting.

We asked whether the naïve state maintenance cABN is consistent with experimental observations of EpiSC resetting. To this end, we exploited GOF18 EpiSCs, which are susceptible to resetting in 2i+LIF in the absence of transgenes (Han *et al*, 2010a). In accordance with the Boolean modelling formalism, we discretised gene expression patterns of the network components for the initial (GOF18 EpiSC) and final (naïve state ESC) states, such that each gene is High/Low in each case (Appendix Fig S1E and Materials and Methods). We defined a set of six constraints based on experimental observations of when EpiSC resetting can or cannot be achieved (Fig 1C, Appendix Fig S1F and Materials and Methods). For example, one constraint specifies that if a given cell has none of the naïve pluripotency factors initially expressed, then 2i+LIF alone is not sufficient to induce the naïve state (Fig 1C, top arrow). In contrast, resetting can be achieved if the initial state is equivalent to GOF18 EpiSCs, which express Oct4, Sox2 and Sall4 (Fig 1C, third arrow from the top). We found that these additional constraints were satisfied by the naïve state maintenance cABN, which suggests that a common network may control both maintenance and induction of naïve pluripotency.

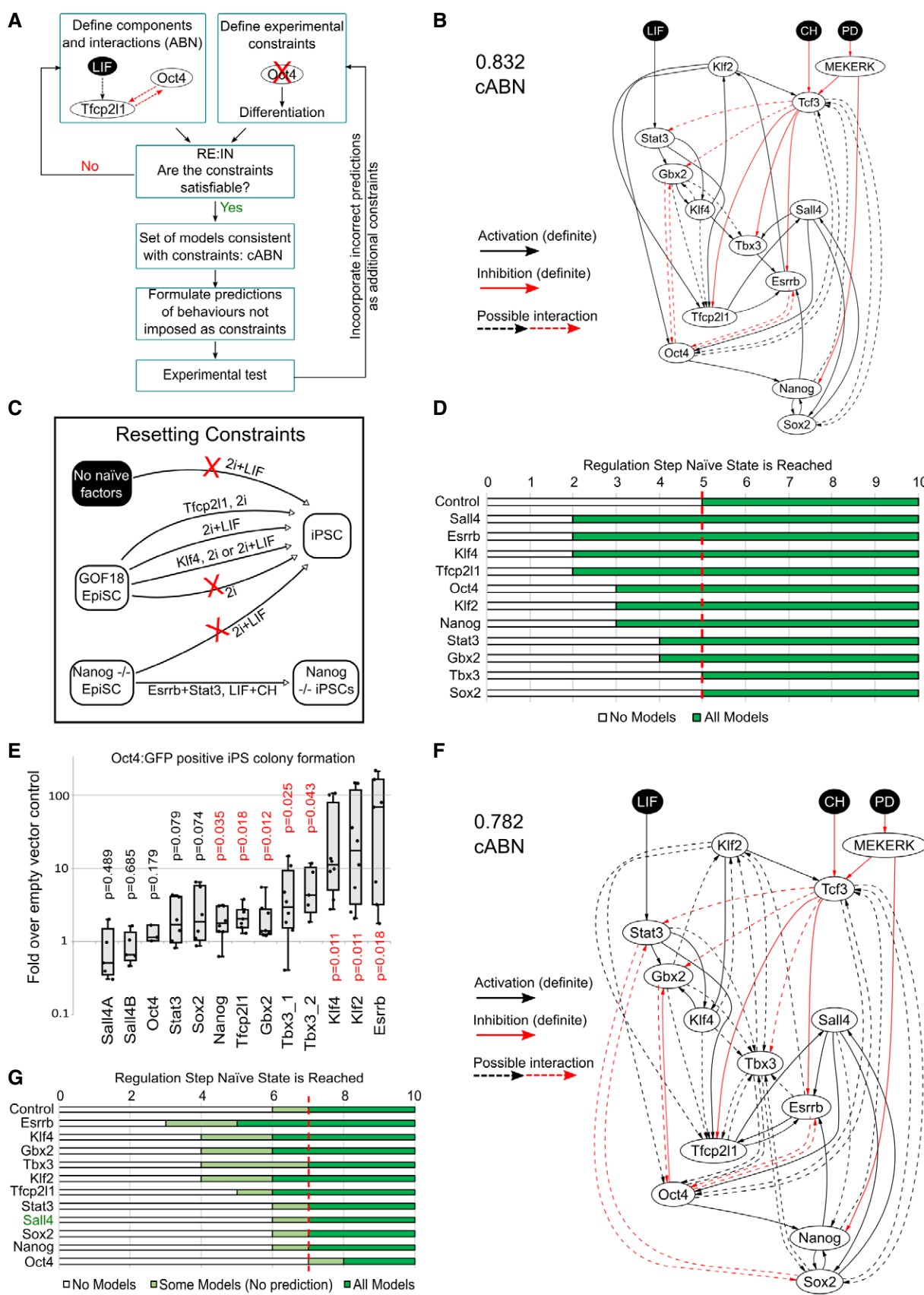

Figure 1.

**Figure 1.   Network models consistent with naïve state maintenance predict the effect of TF forced expression in resetting from primed pluripotency. See also Appendix Figs S1 and S2.**

A   Flowchart describing the methodology. Network components were identified based on functional studies from the literature, and possible interactions between components defined based on pairwise gene expression correlation. A set of experimental results served as constraints. The software RE:IN synthesises all possible interaction networks consistent with the constraints, which is termed the cABN. The cABN is used to formulate predictions to be tested experimentally. Importantly, predictions do not overlap with imposed constraints. If predictions are falsified, the cABN can be further refined by incorporating new experimental results as constraints. The refined cABN is used to generate further predictions.

B   cABN derived from a Pearson coefficient threshold of 0.832, consistent with constraints previously defined for ESC self-renewal (Dunn *et al*, 2014). Solid arrow, required interaction; dashed arrow, possible interaction; black arrow, activation; red arrow, inhibition. There is no regulation hierarchy associated with component positioning.

C   Illustration of EpiSC resetting constraints. See Appendix Fig S1F.

D   Predicted number of regulation steps required for all models to stabilise in the naïve state under forced expression of a single network component. The red dashed line indicates the number of steps required under empty vector control.

E   Fold increase of Oct4-GFP$^+$ colony number over control under forced expression of individual factors. $n \geq 5$, where each dot indicates an independent experiment. Box plots show median, 1$^{st}$ and 3$^{rd}$ quartile values. One-sample Wilcoxon test *P*-values are as indicated, with $P < 0.05$ shown in red.

F   cABN derived from a Pearson coefficient threshold of 0.782.

G   Predictions from the 0.782 cABN. Light green regions indicate where some, but not all, concrete networks allow stable conversion to the naïve state. Sall4 is indicated in green, as this was imposed as a constraint and therefore is not a model prediction.

The number of concrete models in the 0.832 cABN is in the order of $10^5$. As a control, we randomly generated 10,000 models with the same number of components and possible interactions. None of these models could satisfy the entire set of constraints. Indeed, if interactions with a Pearson correlation of at least 0.5 are chosen randomly, the probability of generating the 0.832 ABN is of the order $10^{-31}$. This indicates that the data-driven approach facilitated identification of meaningful interactions between network components, and in practical terms substantially reduced the compute time for subsequent analyses. To test the requirement for each component in the cABN, we explored the consequence of deleting individual TFs from the network and constraints (Materials and Methods). Deleting 8 of the TFs made the initial constraints unsatisfiable. Only removal of Esrrb could be tolerated, but with substantially reduced number and accuracy of predictions. Therefore, the models are highly sensitive to all components of the cABN.

**Prediction of resetting potency for individual network components**

The dynamics of the concrete networks in the cABN were determined by a synchronous update scheme: from a given initial state, each and every component updates its state in response to its upstream regulators at each step (see Materials and Methods). Accordingly, we could examine the sequence of activation of each component along the trajectory towards the naïve state. RE:IN can be used to determine the number of regulation steps required by all models to reach the naïve state. This can be used as a metric to study the resetting process (Materials and Methods).

Spontaneous GOF18 EpiSC resetting can be enhanced by expression of naïve network factors such as Klf2 (Hall *et al*, 2009; Gillich *et al*, 2012; Qiu *et al*, 2015), and such resetting events, measured by reporter activation, often possess faster activation kinetics than control (Gillich *et al*, 2012). The GOF18 EpiSC line contains a transgenic GFP reporter driven by the upstream regulatory region of Pou5f1 (commonly known as Oct4). This transgene does not behave as endogenous Oct4. It is active in ESCs but only in a rare subpopulation of EpiSCs. Therefore, it serendipitously allows the live monitoring of EpiSC to ESC conversion (Han *et al*, 2010a). We hypothesised that enhanced EpiSC resetting upon naïve factor expression may be due to accelerated network activation. We

sought to test this computationally by determining the number of regulation steps required for *all* concrete models of the cABN to stabilise in the naïve state in 2i+LIF, with or without *Klf2* transgene expression. The 0.832 cABN predicted that forced expression of Klf2 in GOF18 EpiSCs results in the network stabilising in the naïve state in only three steps, compared with five steps for transgene-free control (Appendix Fig S2A). Experimentally, we confirmed that transient Klf2 expression induced Oct4-GFP$^+$ colony formation earlier than empty vector control and led to higher colony number throughout 10 days of EpiSC resetting time course (Appendix Fig S2B; Gillich *et al*, 2012). Thereafter, we assumed that the number of Oct4-GFP$^+$ colonies obtained reflected EpiSC resetting dynamics and used this as an experimental output to compare with computational predictions.

We predicted the effect of forced expression of each network component using the 0.832 cABN (Fig 1D). The predictions indicated that expression of all factors except Tbx3 and Sox2 would lead to stabilisation in the naïve state in fewer steps than control, indicating that most network components could enhance EpiSC resetting. For example, when Esrrb is introduced, all concrete models predicted full activation of the naïve network by Step 2, compared to Step 5 for control.

To test these predictions experimentally, we generated expression constructs for each factor by cloning the cDNA into an identical vector backbone and transiently transfected GOF18 EpiSCs 1 day prior to initiating resetting in 2i+LIF. We measured the relative efficiency between different components by the fold increase of Oct4-GFP$^+$ colonies formed at Day 7 over empty vector control (Fig 1E, Appendix Fig S2C). While some factors, such as Sall4 and Oct4, had no significant effect over control, others, notably Esrrb, Klf2 and Klf4, showed a robust enhancement. The computational predictions showed a similar trend to the experimental results, with seven out of eleven cases correctly predicted (Fig 1D, Appendix Fig S2D). Predictions for Tbx3, Stat3 and Oct4 transgene expression were incorrect. Most strikingly, Sall4 was predicted to be one of the most efficient factors, but was found to be the least efficient experimentally.

The iterative nature of our approach (Fig 1A) allows the refinement of the cABN in the light of new experimental results that were predicted incorrectly. We encoded the experimental observation that Sall4 expression was no more efficient than control as an additional

constraint (Materials and Methods). Satisfying the new constraint together with the original set required additional possible interactions, which were identified by lowering the Pearson coefficient threshold (Fig 1F). The new threshold, 0.782, was the highest to define a cABN that satisfied the updated experimental constraints. We then generated a new set of predictions for single factor forced expression (note that forced expression of Sall4 is encoded as a constraint therefore is not used to make a prediction). In each case, we observed a range of steps for which some concrete models predicted stabilisation in the naïve state, while others did not (Fig 1G, light green). However, predictions can only be formulated when all concrete models are in agreement (Fig 1G, dark green). Therefore, forced expression of Esrrb, Klf4, Gbx2, Klf2 or Tfcp2l1 was predicted to be more efficient than control, in agreement with the experimental results shown in Fig 1E (see also Appendix Fig S2D).

For forced expression of Nanog, Tbx3, Stat3 and Sox2, overlap of the light green regions with control prevented definitive predictions (Fig 1G). To resolve this uncertainty, we formally tested *in silico* whether expressing a given factor would be more efficient than control for every concrete model. This resulted in the correct predictions that Nanog was always at least, or more efficient than control, while Stat3, Sox2 and Oct4 were not (Appendix Fig S2D). The strategy did not generate a prediction for Tbx3 because some concrete models generated different kinetics to others.

We extended the test to perform a pairwise comparison of all genes to delineate the relative efficiency of individual factors (Appendix Fig S2E). Predictions could be formulated for 37 out of 55 possible comparisons. Of these, 22 were supported experimentally, while 9 were incorrect. For the remaining 6, the experimental results showed a trend in agreement with the predictions, although without reaching statistical significance due to variability in the naïve colony number between independent experiments. Appendix Fig S2F summarises all significant pairwise comparisons with experimental support.

### Delineating the sequence of network activation

The 0.782 cABN accurately predicted the effect of forced expression of naïve components on EpiSC resetting, which suggests that resetting is not a random process. We therefore asked if resetting occurs via a precise sequence of gene activation, and whether this could also be identified using the cABN. We investigated whether a defined sequence of gene activation was common to all concrete models, or whether individual models transition through unique trajectories. We focussed on those genes expressed at low levels in GOF18 EpiSCs, to enable unequivocal detection of activation over time in population-based measurements.

To predict the sequence of gene activation during EpiSC resetting, we examined the number of regulation steps required for each gene to be permanently activated in 2i+LIF without transgene expression (Fig 2A). The 0.782 cABN predicts that Stat3 and Tfcp2l1 are the first to be activated, at Steps 1 and 2, respectively, while Gbx2, Klf4 and Esrrb are activated last, at Steps 6 and 7. The wide range of step values for permanent Tbx3 activation predicted by different concrete models within the cABN (Fig 2A, light blue region) prevented a definitive prediction in this case.

To test these predictions, we measured the expression of each gene over the EpiSC resetting time course in 2i+LIF for up to 4 days (Fig 2B and C). We defined gene activation to be an upregulated expression level that is statistically significant over EpiSCs. As predicted, Stat3 was significantly induced as early as 2 h after 2i+LIF induction, Tfcp2l1 after 8 h, while Klf4, Esrrb and Tbx3 only became detectable between 48 and 96 h. In contrast to the predictions, Klf2 was significantly increased after only 1 h of 2i+LIF treatment.

Tfcp2l1 and Esrrb are direct targets of the LIF/Stat3 and CH/Tcf3 axes (Martello *et al*, 2012, 2013; Ye *et al*, 2013; Qiu *et al*, 2015). However, even though CH and LIF were applied simultaneously to initiate resetting, Tfcp2l1 and Esrrb displayed distinct activation kinetics. We hypothesised that the local regulation topology of these two components may affect the timing of their activation. We therefore examined all immediate upstream regulators of Tfcp2l1 and Esrrb, and the logical update rules that define the conditions under which each component becomes active (Fig 2D). Tfcp2l1 had six upstream activators, of which Stat3 and Esrrb were definite, and one inhibitor, Tcf3. Esrrb had three definite activators, Sall4, Nanog and Tfcp2l1, as well as a definite and an optional inhibitor. The computational methodology defines a set of alternative update rules, referred to as regulation conditions, that span the possible scenarios under which a target can be activated (Materials and Methods; Yordanov *et al*, 2016). In the same manner in which some possible interactions were found to be required or disallowed when experimental constraints were applied to the ABN, certain regulations conditions were also found to be used or unused in order to satisfy the constraints. We compared the subset of regulation conditions assigned to Tfcp2l1 and Esrrb across all concrete models in the cABN, and one key difference emerged. While Tfcp2l1 required only one of its potential activators (Stat3, Esrrb, Tbx3, Gbx2, Klf2 or Klf4) to activate expression, Esrrb required the presence of all activators (Nanog, Tfcp2l1, Sall4; Fig 2D). Since Stat3 was activated after 1 h in response to 2i+LIF, early activation of Tfcp2l1 could therefore be attributed to Stat3. Esrrb would necessarily only be activated after activation of Tfcp2l1. This local topology analysis therefore provides a network explanation accounting for the rapid

---

**Figure 2. Models predict the sequence of gene activation during resetting to naïve pluripotency.**

A   Model predictions of the number of regulation steps required for permanent activation of each network component. Light blue regions indicate where only some, while dark blue regions indicate that all concrete networks predict that the given gene has permanently activated.

B   Heatmap of average gene expression normalised to β-actin over an EpiSC resetting time course in 2i+LIF. Each row is coloured according to the unique minimum and maximum for that gene. The values shown are average expression of four independent experiments.

C   Gene expression for Stat3, Klf2, Esrrb and Tfcp2l1 during EpiSC resetting relative to established mouse ESCs. β-actin serves as an internal control. Mean ± SEM, *n* = 4 independent experiments. *$P < 0.05$ Student's *t*-test.

D   Left: Local network topology for Tfcp2l1 and Esrrb. Right: Summary of regulation conditions required by Tfcp2l1 and Esrrb in the 0.782 cABN.

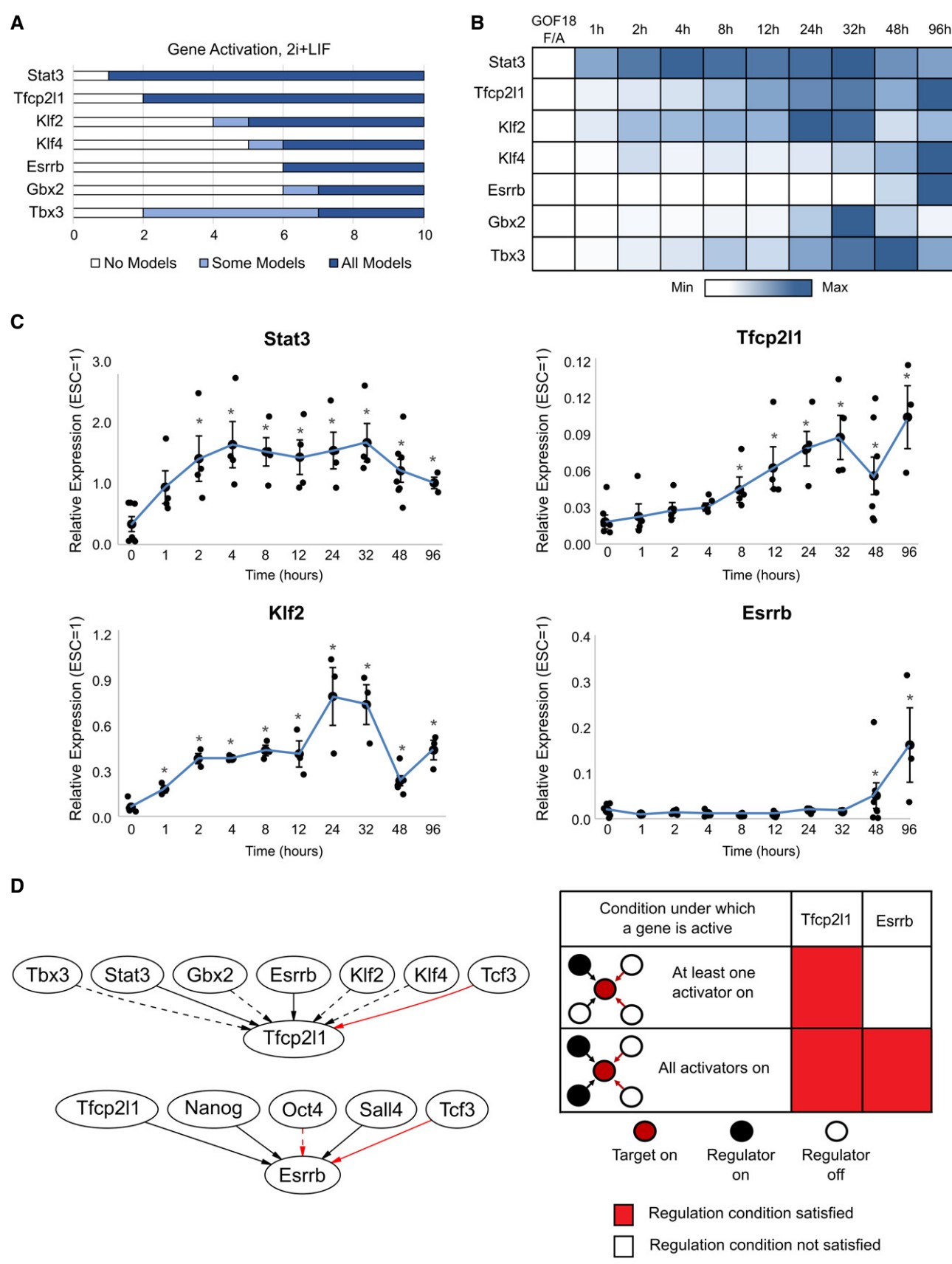

Figure 2.

activation of Tfcp2l1 (8 h, Fig 2B) and the delayed activation of Esrrb (48 h).

## Combinations of factors can enhance EpiSC resetting

Earlier studies have shown that forced expression of a combination of factors can synergistically enhance resetting efficiency (Yang *et al*, 2010; Gillich *et al*, 2012; Qiu *et al*, 2015). Our computational approach enabled us to investigate the effect of factor combinations in a systematic manner. We focused on those factors found to be potent inducers when expressed individually: Klf4, Klf2, Esrrb, Tbx3 and Tfcp2l1 (Fig 1E). Six out of seven combinations were predicted to reduce the number of regulation steps required to induce and stabilise the naïve state (Fig 3A, left). In the case of Esrrb/Tfcp2l1 dual expression, no enhancement beyond single factors was predicted.

We tested these combinations experimentally by transient transfection of the factors singly or combined. The number of reset Oct4-GFP$^+$ colonies was scored at Day 7, and resetting efficiency was calculated based on fold increase over empty vector control. The resetting efficiency of dual factor transfection was compared to individual factors alone to determine the combinatorial effect. Six out of seven experimental results were consistent with computational predictions (Fig 3A, right). Five combinations (Esrrb/Klf2, Esrrb/Klf4, Esrrb/Tbx3, Klf4/Tbx3, Klf2/Tbx3) yielded synergistic enhancement, while two combinations (Esrrb/Tfcp2l1 and Klf2/Klf4) showed no greater effect than the single factors (Fig 3A, right, 3B). These results demonstrate that the logic encoded within our data-constrained set of models is sufficient to predict synergistic or non-additive behaviour of factor combinations.

Since dual expression of Esrrb/Tbx3 and Esrrb/Klf4 dramatically enhanced EpiSC resetting (Fig 3B, right), we utilised these combinations to explore resetting dynamics in detail. We generated *piggyBac* vectors harbouring doxycycline (DOX)-inducible Esrrb-T2A-Klf4-IRES-Venus and Esrrb-T2A-Tbx3-IRES-Venus constructs. We delivered the transgenes into GOF18 EpiSCs together with a separate rtTA construct (Fig 3C). The presence of Venus$^+$ cells upon DOX treatment confirmed induction of transgene expression. As a control, we used an empty vector carrying only the DOX responsive element and IRES-Venus. To assay resetting potency, we transferred EpiSCs to 2i+LIF in the absence or presence of DOX (0.2 μg/ml) for 48 h and continued resetting in 2i+LIF only for an additional 4 days before scoring Oct4-GFP$^+$ colonies (Fig 3D). Cells transfected with the empty vector with or without DOX, or with expression constructs in the absence of DOX, showed spontaneous resetting at low frequency, as expected (Fig 3E, top). In contrast, both factor

combinations in response to DOX yielded robust Oct4-GFP activation. There were too many GFP$^+$ colonies to score accurately, and therefore, we quantified the GFP signal intensity of randomly selected fields (Fig 3E, bottom). This analysis demonstrated that DOX induction led to a 9- to 16-fold increase in Oct4-GFP expression for Esrrb/Tbx3 and Esrrb/Klf4 expression respectively.

To examine EpiSC resetting kinetics functionally, we replated cells after 2, 4, 6 or 8 days (Fig 3D) at clonal density and scored the number of emergent alkaline phosphatase (AP)-positive colonies. In the absence of DOX, both the empty vector and dual expression transfectants exhibited gradual accumulation of a few colonies. With DOX induction, however, dual factor transfectants displayed rapid production of numerous AP$^+$ colonies, commencing as early as Day 2 and peaking at Day 6 (Fig 3F).

To investigate whether the effect of these combined factors extended to other EpiSC resetting systems, we expressed these combinations in an independent EpiSC line, OEC2 (Appendix Fig S3A), which carries an Oct4-GFP transgene and the chimeric LIF receptor GY118 (Yang *et al*, 2010). Resetting does not occur in this cell line in 2i+LIF alone. Similar to GOF18 EpiSCs, we found robust induction of Oct4-GFP$^+$ colony formation with DOX treatment and could observe resetting to the naïve state with only 24 h of DOX induction (Appendix Fig S3B–E).

## Delineating the sequence of network activation under dual factor expression

We next used the 0.782 cABN to investigate the sequence of gene activation that occurs upon dual factor expression. Predictions were generated for the number of regulation steps required for each component to be permanently activated (Fig 4A and B top, Appendix Fig S4), and compared with experimental results (Fig 4A and B bottom). To generate the experimental results, we measured network component expression of DOX-inducible GOF18 EpiSCs carrying the empty vector, Esrrb-T2A-Tbx3 or Esrrb-T2A-Klf4 constructs, and treated with 2i+LIF in the presence or absence of DOX.

Under DOX treatment, Esrrb-T2A-Tbx3 transfectants showed a more robust activation of endogenous Klf2 and Klf4 at Day 4 relative to non-induced cells, consistent with the prediction that these genes should be activated earlier (Fig 4A). Stat3 upregulation was not accelerated, also as predicted. However, in the case of Tfcp2l1, we detected enhanced activation that was not predicted. For Esrrb-T2A-Klf4 expression, we observed accelerated activation of Klf2 and Tbx3 at Day 4 compared to control, consistent with predictions (Fig 4B). Again, we observed enhanced activation of Tfcp2l1 that

---

**Figure 3. Combinations of potent factors enhance resetting by accelerating network activation. See also Appendix Fig S4.** ▶

A Left: Comparison of the number of steps required for all concrete networks to stabilise in the naïve state under single and dual factor expression. Right: Experimental results showing the fold increase in colony number over empty vector control under single and dual factor expression. Y = Yes, N = No, *incorrect prediction.

B Predictions and experimental validation of examples of synergistic and non-additive factor combinations. Experimental measurement of fold increase over empty vector control of Oct4-GFP$^+$ colony numbers. Bars indicate the mean of 2 independent experiments, shown as dots.

C Scheme for DOX-inducible constructs used for dual factor expression.

D Experimental set-up for the functional characterisation of Esrrb-T2A-Klf4 or Esrrb-T2A-Tbx3 forced expression in EpiSC resetting.

E Representative confocal images (top) and quantification (bottom) of Oct4-GFP reporter mean intensity. The indicated cell lines were treated with DOX for the first 2 days and imaged at Day 6. Mean ± SD of 9 technical replicates. One representative experiment of two is shown. Scale bars = 300 μm.

F Representative alkaline phosphatase (AP) staining images (left) and quantification (right) of AP$^+$ colonies after clonal replating, as described in panel. Mean ± SEM, *n* = 3 independent experiments.

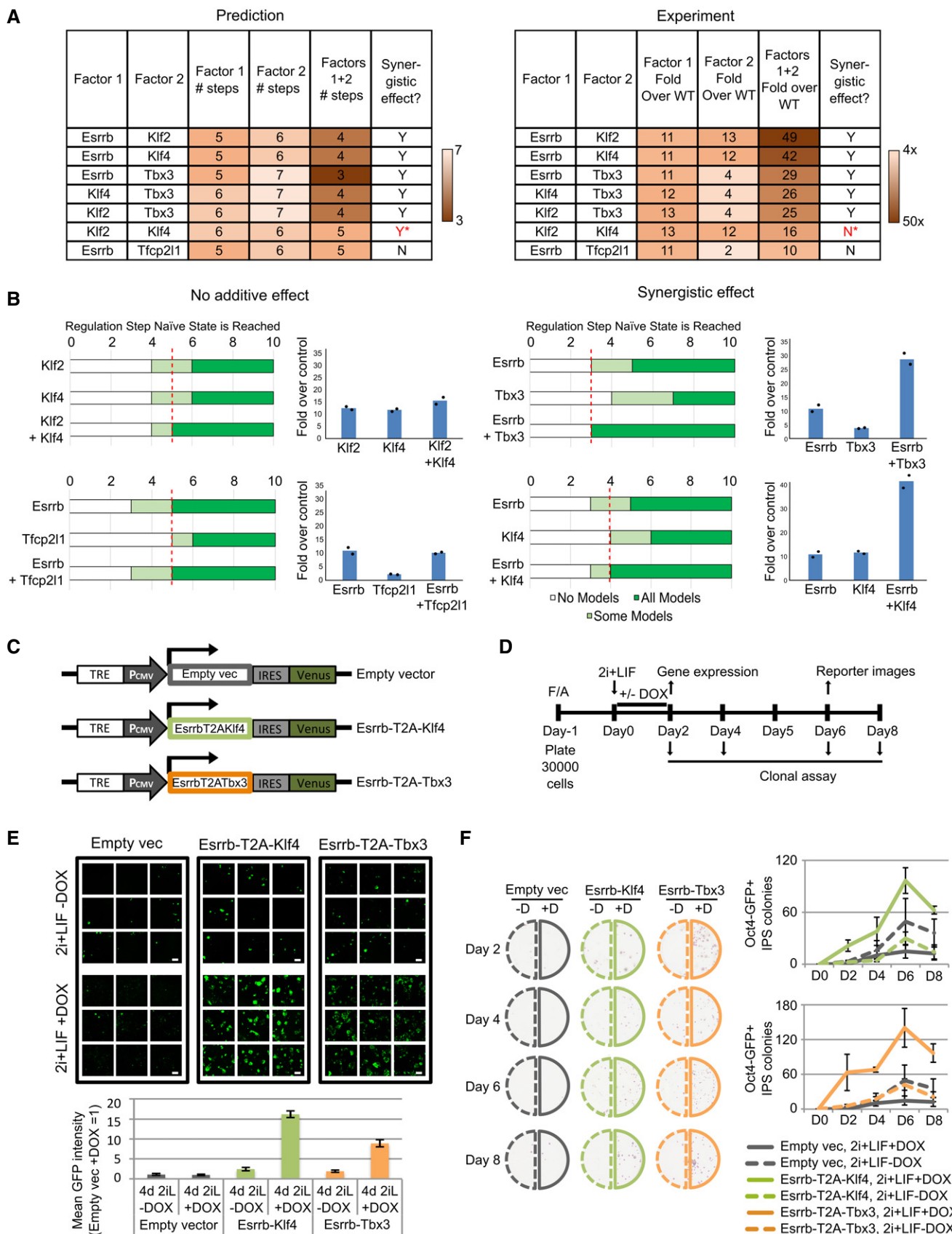

**Figure 3.**

was not predicted, while Stat3 showed no enhancement over control. Importantly, the sequence of gene activation was independently validated in OEC2 EpiSCs (Appendix Fig S3F). Of note, the level of Klf2 activation relative to ESCs is much lower in OEC2 (Appendix Fig S3F) compared to GOF18 EpiSCs (Fig 4A and B).

EpiSC resetting is typically an inefficient and asynchronous process (Appendix Fig S2B), limited by both technical and biological variability. Consequently, analysis of population-based measurements could mask the precise sequence of gene activation in productive resetting. Since the inducible Esrrb-T2A-Klf4 expression system significantly enhanced EpiSC resetting, this enabled us to capture gene activation kinetics at single-cell resolution. Examining gene expression of different components within the same cell along the EpiSC resetting trajectory should allow reliable characterisation of the sequence of network activation. To achieve this, we sorted individual cells after 2 and 4 days resetting in 2i+LIF with DOX treatment (Fig 4C left). We conducted single-cell gene expression profiling by RT–qPCR of Day 2 Venus/GFP Low and High and Day 4 Venus/GFP High cells that were clonogenic in 2i+LIF upon replating, i.e. those that had undergone productive reprogramming (Fig 4C, right). As controls, we included established mouse RGd2 ESCs (Kalkan *et al*, 2017) and un-induced parental EpiSCs. We profiled selected genes that were differentially expressed between naïve ESCs and primed EpiSCs along with the core pluripotency factors, Oct4 and Sox2 (Fig 4D and E, Table EV1). Robust activation of naïve ESC-associated genes was observed in Day 4 Venus/GFP High cells. Some genes, such as Oct4 and Nr5a2, showed even higher expression in Day 4 Venus/GFP High cells than in ESCs (Fig 4D), possibly due to the perduring expression of Esrrb and Klf4. Oct4 and Sox2 were reduced in many, but not all, Day 2 Low cells, but were robustly expressed in some Day 2 High and all Day 4 High samples (Fig 4E). EpiSC-enriched genes that are also expressed at low levels in naïve ESCs, such as Otx2, Utf1 and Pim2, were downregulated at Day 2. However, some Day 4 High cells re-acquired expression of those genes associated with early transition from naïve pluripotency (Acampora *et al*, 2016; Kalkan *et al*, 2017). In established reset clones, however, their expression levels were similar to ESCs (Appendix Fig S5A). Naïve pluripotency in such reset clones was confirmed functionally by generation of multiple high-grade live-born chimera (Appendix Fig S5B). Overall, the single-cell transcriptional analysis further validated the robust, stable activation of the naïve network after 4 days of Esrrb-T2A-Klf4 expression.

## Clustering of single-cell gene expression profile reveals an EpiSC resetting trajectory

We explored the sequence of gene activation at single-cell resolution by examining the proportion of cells displaying expression of individual genes at different stages of resetting to test whether these data were consistent with predictions. For example, the 0.782 cABN predicted that Klf2 would always be active before Tbx3, from which it follows that upregulation of Tbx3 should not occur in the absence of Klf2.

To test these predictions, we first discretised the data by *k*-means clustering (Materials and Methods) and calculated the proportion of cells at each resetting stage exhibiting the four expression patterns: Klf2/Tbx3 both Low; Klf2 High and Tbx3 Low; Klf2/Tbx3 both High; and Klf2 Low and Tbx3 High (Fig 5, top row). The majority of cells at Day 2 were Klf2/Tbx3 double Low, while such cells were not found at Day 4. By Day 4, the majority of cells were Klf2/Tbx3 double High, as for ESCs. We observed that the proportion of Klf2 High and Tbx3 Low cells was highest at Day 2 and subsided at Day 4 and in the established ESCs. Klf2 and Tbx3 High cells were mostly present at Day 4 and ESCs. This indicates that Klf2 activation precedes Tbx3. Only a negligible fraction of Klf2 Low and Tbx3 High cells was observed in Day 4. Since similar fraction of these cells was observed in ESCs, this indicates that they most likely reflect transcriptional fluctuation or heterogeneity in the naïve state. These results are consistent with the prediction that Klf2 precedes Tbx3 activation during resetting upon Esrrb-T2A-Klf4 expression. Similarly, the cABN accurately predicted that Klf2 activation precedes Gbx2, and sustained expression of Sox2 is prerequisite to Tfcp2l1 activation (Fig 5, middle and bottom panels). As an independent approach, we performed hierarchical clustering using the SPADE algorithm to visualise the kinetics of gene activation (Anchang *et al*, 2016; Qiu *et al*, 2011; Appendix Fig S5C and D). This analysis confirmed our observations and also allowed us to place factors that are not in the naïve network on the resetting activation timescale. For example, Nr5a2, a known resetting enhancing factor (Guo & Smith, 2010), activates in a similar pattern to Klf2.

## Identifying required components for naïve network activation

We next investigated whether loss of specific network factors would block naïve network activation. We used the 0.782 cABN to predict

---

**Figure 4.  Co-expression of factors activated late in EpiSC resetting increases pluripotency marker expression and significantly reduces the resetting time scale. See also Appendix Figs S3 and S5, Table EV1.**

A   Top: Predictions of the number of regulation steps required for full activation of the indicated gene under control or dual expression of Esrrb and Tbx3 (+E/T). Bottom: Gene expression of EpiSCs harbouring empty vector (grey) or Esrrb/Tbx3 (orange), captured at D0 (F/A), D2 and D4 (as described in Fig 3D). Dashed black line: expression levels in ESCs maintained in 2i+LIF. Data normalised to empty vector cultures in F/A. Gapdh serves as an internal control. Mean ± SEM, *n* = 3 independent experiments.

B   As for panel (A), comparison of control with dual expression of Esrrb and Klf4 (+E/K, green in bottom plot). Mean ± SEM, *n* = 3 independent experiments for bottom qRT–PCR panels.

C   Left: Flow cytometry profiles of the resetting progression of EpiSCs stably transfected with the Esrrb-T2A-Klf4 construct and cultured in 2i+LIF with DOX for 2 and 4 days, with the indicated fraction of cells sorted for colony formation assay. Since the Venus reporter is under the control of a DOX responsive element, and the emission spectra of Venus and GFP fluorescence overlap, the Oct4-GFP reporter could not be fully distinguished from Venus expression. Right: Number of AP[+] colonies formed from 250 sorted cells from indicated fractions. Data points represent two technical replicates of one out of two independent experiments.

D   Heatmap of single-cell expression measured by qRT–PCR of major ESC and EpiSC markers in un-induced EpiSCs (black), established ESCs (red), Day 2 High/Low (dark and light blue) and Day 4 High cells (green).

E   Scatterplots of the relative expression of pluripotency and transitional markers. Red bar corresponds to median gene expression, and each dot represents a single cell.

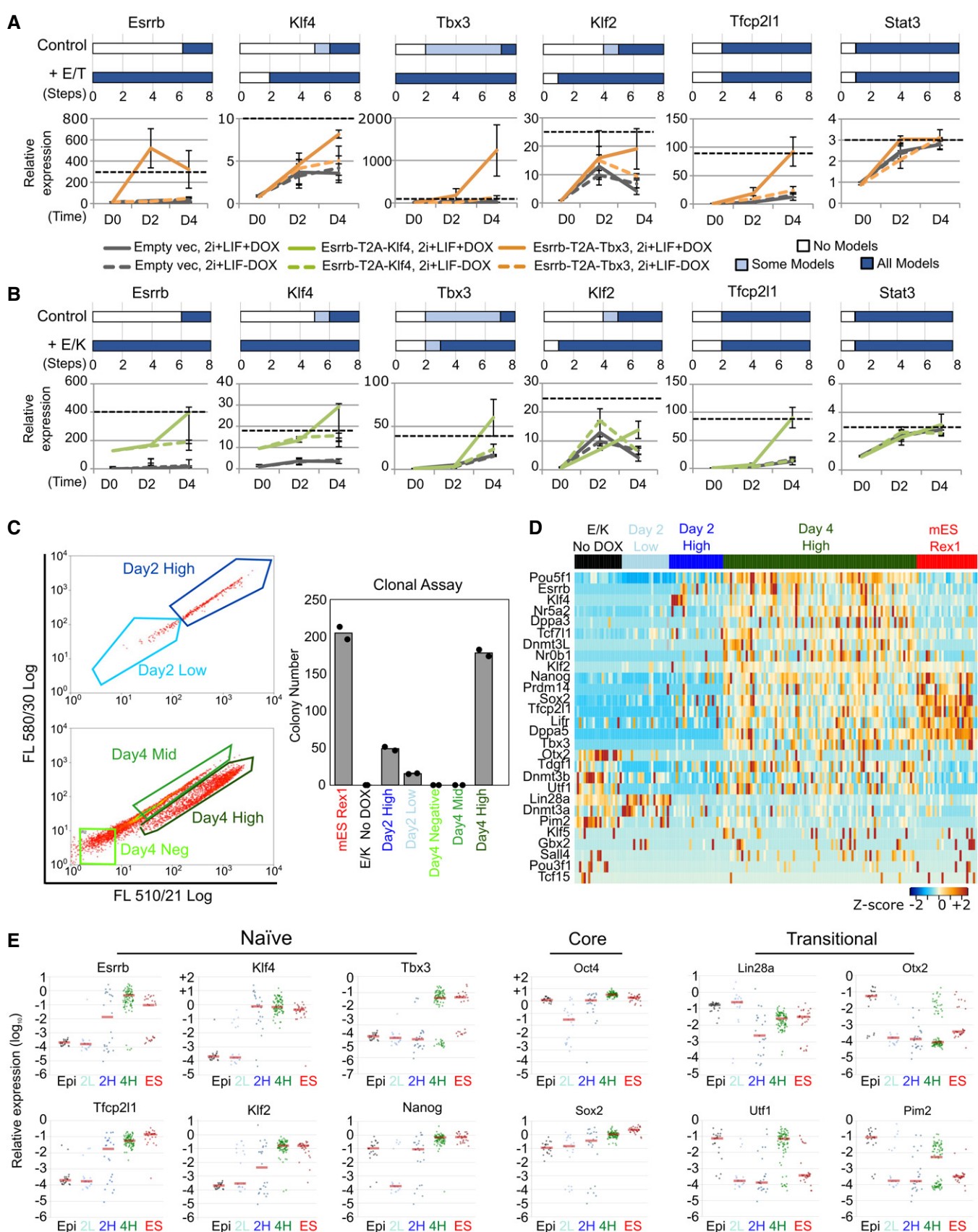

**Figure 4.**

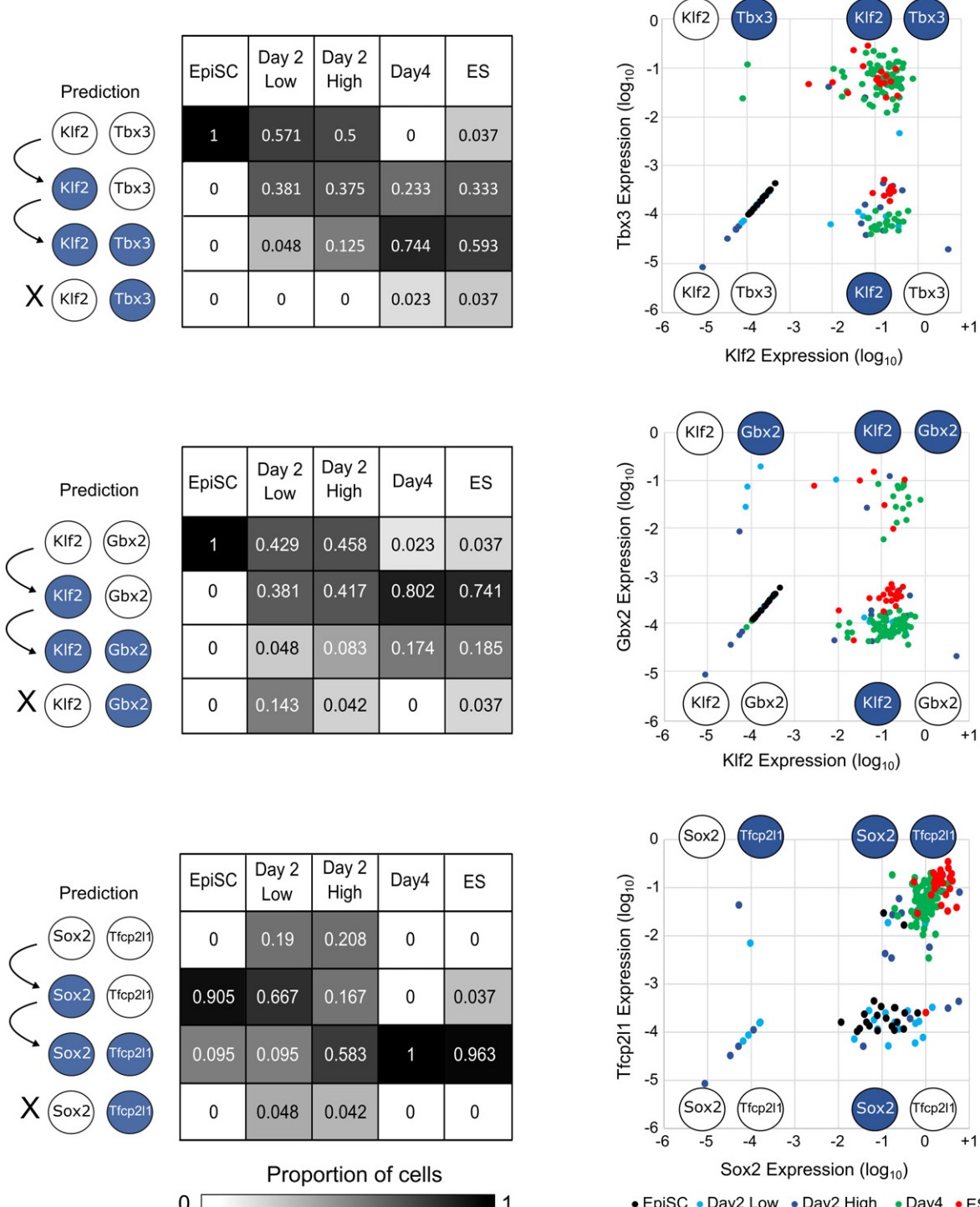

**Figure 5. Single-cell gene expression profiles recapitulate the predicted sequence of gene activation.**

Left: 0.782 cABN predictions of the sequence of gene activation between gene pairs (white, Low; blue, High) along the resetting trajectory, compared to single-cell gene expression measured by RT–qPCR. Each table summarises the percentage of single-cells at the indicated stage of resetting (columns) that have the indicated expression state (rows). Right: Scatterplots showing single-cell coordinates based on the expression of the gene pair.

network components required for EpiSC resetting by investigating whether the network could permanently stabilise in the naïve state in the absence of each factor (Fig 6A). The 0.782 cABN predicted

that two factors, Esrrb and Gbx2, are dispensable for EpiSC reset-ting, while the remaining factors are required. In the case of Tbx3, no prediction could be formulated.

                                                  

To test these predictions, we transfected GOF18 EpiSCs with siRNAs against individual network factors. EpiSC resetting was initiated 24 h post-transfection by switching from F/A to 2i+LIF, and Oct4-GFP$^+$ colonies were scored at Day 6 (Fig 6B). Experimental results confirmed that Gbx2 is dispensable for resetting. Furthermore, the requirements for Oct4, Sall4, Sox2, Stat3 and Klf2 were accurately predicted. Knockdown of Esrrb and Tbx3 reduced but did not abolish colony formation. Overall, 6 out of 9 predictions were consistent with experimental results.

The experimental results revealed distinct resetting behaviour upon Klf2 or Klf4 depletion. Both were predicted to be required, yet Klf4 knockdown did not eliminate colony formation, while Klf2 was found to be essential (Fig 6B). This experimental result was counterintuitive as well as not predicted. Klf4 and Klf2 show at least partially redundant function in ESC self-renewal (Jiang *et al*, 2008), and both were potent resetting inducers when expressed in GOF18 EpiSCs (Fig 1E). To confirm the result, we generated *Klf2* and *Klf4* knockout (KO) GOF18 EpiSCs by deleting the largest coding exons using CRISPR/Cas9 (Appendix Fig S6A and B). Resetting in 2i+LIF using two independent KO EpiSC clones confirmed the knockdown results: *Klf4* KO EpiSCs generated Oct4-GFP$^+$ colonies as efficiently as wild-type control, while *Klf2* KO EpiSCs yielded no Oct4-GFP$^+$ colonies (Fig 6C). This observation was further validated using an independent EpiSC line in which resetting is driven by hyperactivation of Stat3 (Appendix Fig S6C).

To investigate the consequence of Klf2 loss for network activation, we examined the expression of network components over the resetting time course for up to 4 days (Fig 6D). Wild type (WT) and *Klf2* KO EpiSCs showed similar patterns of expression for Oct4, Sox2 and Sall4. For up to 8 hours of resetting, *Klf2* KO cells behaved similarly to WT. However, *Klf2* KO cells failed to elevate the expression of Nanog and Tfcp2l1 at later time points. Factors activated after 2 days of resetting, such as Esrrb, Klf4 and Tbx3, failed to be activated in *Klf2* KO cells. Taken together, these data suggest that in the absence of Klf2, EpiSCs can respond to 2i+LIF to initiate resetting, but this response is not sustained. Of note, EpiSC markers—Pou3f1, Otx2, Fgf5—were sharply downregulated in both WT and *Klf2* KO cells (Fig 6D, middle), suggesting that Klf2 is not involved in the dissolution of EpiSC identity. Furthermore, in both WT and *Klf2* KO

EpiSC resetting we observed similar upregulation at the population level of lineage-specific genes, such as Sox1 and Pax6 (ectoderm), T/Bra (primitive streak), Flk1 (mesoderm) and Pdgfrα (endoderm; Fig 6D bottom). This suggests that Klf2 does not exert a lineage repression function during resetting. In addition, *Klf2* deletion in ESCs did not affect multi-lineage differentiation (Appendix Fig S6D).

We next asked whether forced expression of individual network factors could compensate for the loss of Klf2. To this end, we transiently expressed all individual factors and found that only Klf2 and Klf4 could rescue the *Klf2* KO phenotype (Fig 6E). These results indicate that Klf2 is specifically required for initiating resetting in EpiSCs. Rescue by forced expression of Klf4 suggests that the two factors are functionally equivalent. Differences in the activation kinetics of the two factors during resetting (Fig 2B) underlie the requirement for Klf2 and dispensability of Klf4 (see also Discussion).

We picked and expanded individual *Klf2* KO reset clones obtained via transient Klf2 expression at Day 7, and confirmed they were free of integration of *Klf2* transgene by genomic PCR (Appendix Fig S6E) and lack of Klf2 expression (Appendix Fig S6F). We quantified gene expression for naïve network factors in these lines and found that Oct4, Tbx3, Tfcp2l1 and Klf4 levels were comparable to control, while Sall4, Gbx2, Sox2, Stat3 and Nanog were elevated (Appendix Fig S6F). Only in the case of Esrrb was gene expression lower in *Klf2* KO iPSCs than in control. Despite these differences, *Klf2* KO naïve cells showed sustained self-renewal in 2i+LIF. Therefore, Klf2 is dispensable for maintenance in 2i+LIF once naïve pluripotency has been attained, consistent with previous reports for ESC propagation (Yeo *et al*, 2014).

In the light of the unexpected finding that Klf2 was specifically required for EpiSC resetting, we investigated the relevance of other network components for resetting versus maintenance of naïve pluripotency. Predictions were generated and tested by siRNA transfection in self-renewing ESCs followed by clonal assay (Dunn *et al*, 2014). Stat3 and Klf2 emerged as specifically required for resetting. Depletion of Tbx3, Esrrb, Nanog and Sall4 also reduced EpiSC resetting frequency but had little effect on naïve state maintenance (Fig 6F). Klf4, Tfcp2l1 and Gbx2 appear dispensable for both maintenance and resetting, while Oct4 and Sox2 are essential

**Figure 6.  Klf2 and Stat3 are required for EpiSC resetting, but not for naïve state maintenance. See also Appendix Figs S6 and S7.**

A   0.782 cABN predictions of essential or dispensable factors for EpiSC resetting, compared with the experiment results shown in panel (B).

B   siRNA knockdown effects measured by Oct4-GFP$^+$ colony formation at Day 6 of resetting. *n* = 4 independent experiments. Box plots indicate 1$^{st}$, 3$^{rd}$ quartile and median. *$P$ < 0.05 Student's *t*-test; n.s. = not significant.

C   Left, resetting capacity of *Klf2* and *Klf4* KO EpiSCs measured by Oct4-GFP$^+$ colony formation at Day 6 of resetting. Data points represent two independent experiments. Right, representative fluorescent and bright field images of wild-type and *Klf2* KO EpiSCs at Day 6 of resetting in 2i+LIF. Scale bars = 100 μm.

D   Expression of naïve pluripotency, transition and somatic lineage markers in wild-type and *Klf2* KO EpiSCs during a resetting time course in 2i+LIF. Expression is normalised to wild-type EpiSCs in A/F, and β-actin was used as internal control. The values shown correspond to the average expression of three technical replicates from one representative experiment out of two.

E   Rescue of *Klf2* KO EpiSC resetting by forced expression of individual network components measured by Oct4-GFP$^+$ colony formation at Day 6 of resetting in 2i+LIF. Data points represent two independent experiments.

F   Comparison between the effect of single factor knockdowns on ESC maintenance (*n* = 4 independent experiments) and EpiSC resetting using experimental results.

G   EpiSC resetting in 2i+LIF measured by Oct4-GFP$^+$ colony formation after Stat3 siRNA in wild-type EpiSCs (left), or *Klf4* KO EpiSCs transfected with Tfcp2l1 and Gbx2 siRNAs. *n* = 3 independent experiments in wild-type cells (box plots indicate min, median, max), *n* = 2 for *Klf4*$^{-/-}$ cells (box plots indicate min, mean, max).

H   EpiSC resetting in 2i+LIF measured by Oct4-GFP$^+$ colony formation of Stat3 knockdown EpiSCs transiently transfected with Tfcp2l1, Gbx2 and Klf4. *n* = 4 independent experiments: Student's *t*-test, *P*-value indicated on plot. Box plots indicate min, median, max. See also Appendix Fig S7C.

I   The 0.717 cABN used to illustrate the kinetics of EpiSC resetting as determined experimentally. Left: Genes coloured according to the order of activation during resetting in 2i+LIF. Right: Genes coloured according to their potency in enhancing the efficiency of resetting. TFs with a green border are the common factors required for ESC self-renewal and EpiSC resetting. See also Appendix Fig S6A.

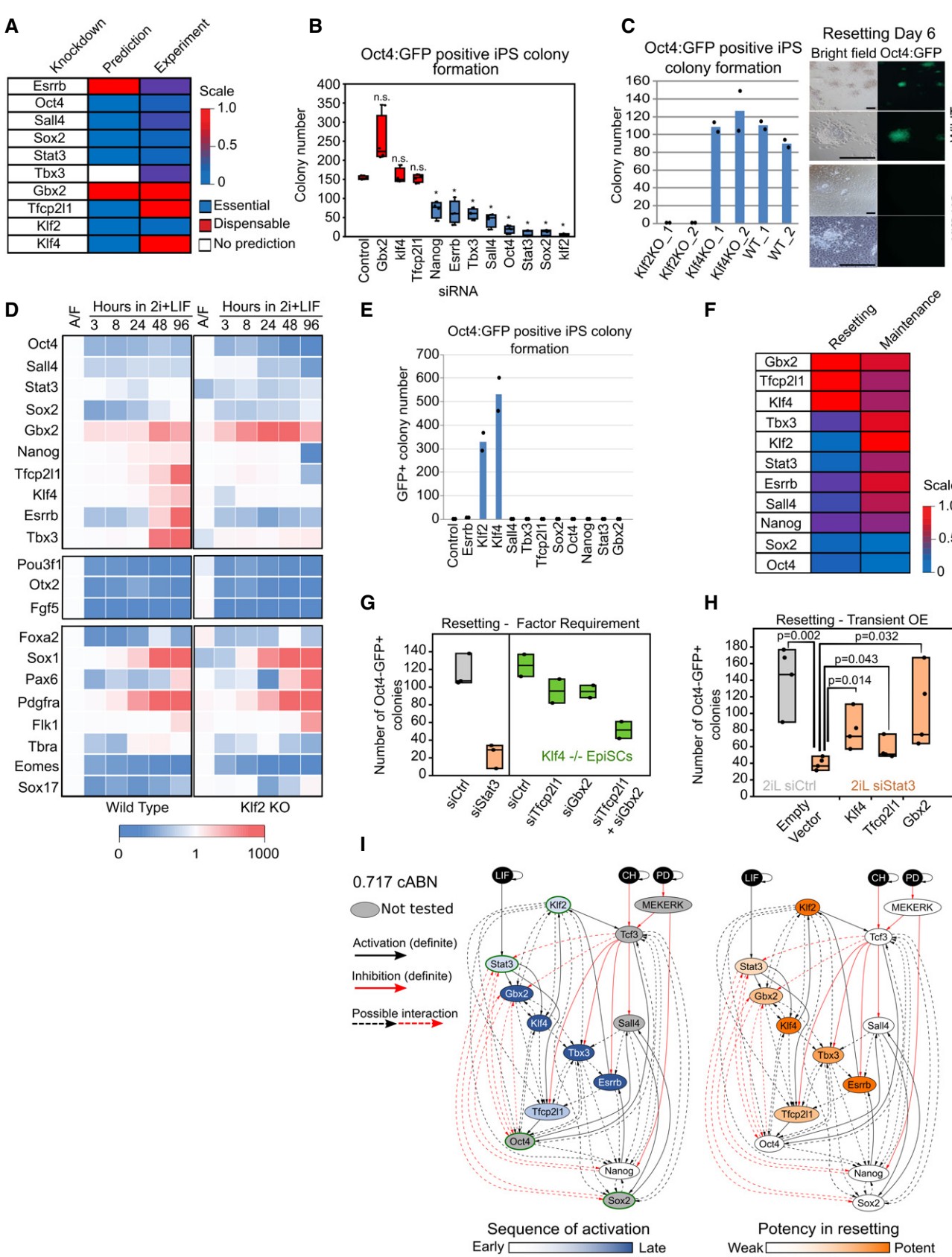

**Figure 6.**

to both (Fig 6F). These results indicate that EpiSC resetting and naïve state maintenance display different sensitivity to network components, and such differences were correctly identified by the 0.782 cABN.

We further investigated the specific requirement for Stat3 in EpiSC resetting. Gbx2, Klf4 and Tfcp2l1 are the direct downstream effectors of Stat3 (Niwa *et al*, 2009; Martello *et al*, 2013; Tai & Ying, 2013). We first examined activation of these TFs in the absence of LIF, or upon Stat3 knockdown. Induction of Tfcp2l1 and Gbx2 was significantly reduced at 24 h in both conditions (Appendix Fig S7A). Later induction of Klf4 was also reduced following Stat3 depletion (Appendix Fig S7B). To examine the functional contribution of these factors downstream of Stat3, we conducted knockdown and rescue experiments. Depletion of Tfcp2l1, Gbx2 or Klf4 either individually or in dual combinations does not inhibit GOF18 EpiSC resetting (Fig 6A and G). However, combined loss of all three factors significantly reduced resetting efficiency (Fig 6G) to levels comparable to Stat3 knockdown. In contrast, forced expression of individual factors rescued the effect of Stat3 knockdown (Fig 6H, Appendix Fig S7C). Taken together, we conclude that Tfcp2l1, Gbx2 and Klf4 are individually dispensable, but in combination they mediate the effect of LIF/Stat3 signalling.

The dispensability of Klf4 and Tfcp2l1 and partial requirement for Esrrb were not correctly predicted by our models (Fig 6A). By including new constraints for the effect of Klf4 and Tfcp2l1 knockdown, we could derive a cABN that was fully consistent with the siRNA resetting phenotypes. Figure 6I shows the refined cABN, defined by a Pearson threshold of 0.717, and also highlights the experimentally observed kinetics of gene activation during EpiSC resetting alongside the potency of individual factors in accelerating the resetting dynamics. Interestingly, the rich set of behaviours we have explored could be explained by as few as 32 interactions between all network components in one "minimal" network topology (Appendix Fig S8A). We characterised both required and disallowed interactions in the 0.717 cABN against CHIP-sequencing data generated from self-renewing mouse ESCs (Sanchez-Castillo *et al*, 2015) and found that 90.91% were supported (Table EV4). This suggests that a large fraction of the interactions may be direct.

## A single biological program governs maintenance and induction of naïve pluripotency

Developmentally distant somatic cell types such as mouse embryonic fibroblasts (MEFs) can be reprogrammed to naïve pluripotency with naïve factor combinations (Takahashi & Yamanaka, 2006). We therefore asked whether MEF reprogramming could also be predicted with the 0.717 cABN. We first surveyed the literature for those factor combinations present in our network that have been used to reprogram MEFs. Without encoding any additional constraints, the 0.717 cABN accurately computed the capacity for successful production of iPSCs for 7 combinations from the literature (Takahashi & Yamanaka, 2006; Nakagawa *et al*, 2007; Silva *et al*, 2008; Feng *et al*, 2009; Buganim *et al*, 2012; Tang *et al*, 2012; Fig 7A). In each case, we assumed a starting state in which all components are inactive, save those factors in the reprogramming cocktail. We next investigated the effect of adding single factors to OSKM in a systematic manner by comparing the number of regulation steps required to stabilise in the naïve state in LIF (Fig 7B). Experimentally, we conducted OSKM reprogramming of primary MEFs with a Nanog-GFP knock-in reporter (TNGA; Chambers *et al*, 2007). Reprogramming was induced by LIF addition in the presence of Vitamin C and Alk inhibitor (O'Malley *et al*, 2013), and Nanog-GFP$^+$ colonies were scored at Day 12 (Fig 7B, right). The 0.717 cABN accurately predicted that the addition of Nanog, Tbx3 and Esrrb would enhance reprogramming efficiency in presence of LIF (Takahashi & Yamanaka, 2006; Nakagawa *et al*, 2007; Silva *et al*, 2008; Feng *et al*, 2009), while Sall4, Gbx2, Klf2 and Tfcp2l1 would have no additive effect.

We also explored the effect of 2i on somatic cell reprogramming. We conducted reprogramming as before, but from Day 6, 2i was supplemented until day 12 when Nanog-GFP$^+$ iPSC colonies were scored. 2i addition enhanced MEF reprogramming compared to LIF alone (Fig 7C). However, LIF is critical to reprogramming over OSKM alone -driven reprogramming irrespective of 2i (Appendix Fig S7D), in agreement with model predictions and previous observations (Silva *et al*, 2008). In 2i+LIF, 3 out of 4 predictions of enhanced reprogramming over OSKM alone proved correct (Nanog, Tbx3 and Esrrb, but not Sall4; Fig 7C). Taken together, these results

---

**Figure 7. A common gene regulatory program governs naïve state maintenance, EpiSC resetting and somatic cell reprogramming.**

A   0.717 cABN predictions compared with published data on gene combinations that do (dark green) or do not (white) enable MEF reprogramming.

B   Comparison of predictions (left) and experimental outcome (right) for the potency of additional network factors in OSKM-driven MEF reprogramming in LIF containing medium. *n* = 4 independent experiments, except Klf2 and Nanog where *n* = 2 independent experiments. *P < 0.05 Student's *t*-test. n.s. = not significant. Red dashed lines indicate empty vector + OSKM level. Box plots indicate min, median, max.

C   Comparison of predictions (left) and experimental outcome (right) for the potency of additional network factors in OSKM-driven MEF reprogramming in 2i+LIF. *n* = 4 independent experiments, except Klf2 and Nanog where *n* = 2 independent experiments. *P < 0.05 Student's *t*-test. n.s. = not significant. Empty vector + OSKM reprogramming in LIF ("control no 2i") was included as a control for the effect of 2i addition. Red dashed lines indicate empty vector + OSKM control level. Box plots indicate min, median, max. The red asterisk indicates that the effect of Sall4 was incorrectly predicted.

D   Recapitulation of the gene activation kinetics during MEF reprogramming. Top, the number of regulation steps required for permanent activation of the indicated gene. Tfcp2l1 and Sall4 are found to activate earlier than Nanog and Esrrb. Bottom, gene expression measured from sorted populations of reprogramming intermediates from O'Malley *et al* (2013).

E   Delineation of gene activation at single-cell level. Top, experimental scheme used in Buganim *et al* (2012) for the isolation of reprogramming intermediates profiled by single-cell RT–qPCR. Bottom, comparison of the predictions of the sequence of gene activation between gene pairs (left) along the reprogramming trajectory in OSKM+LIF, compared with experimental measurements from Buganim *et al* (2012). Each table shows the percentage of single cells at the indicated stage of reprogramming (column) that have the indicated expression state of the gene pair considered (row).

F   Summary of the predictive accuracy of the three cABNs progressively refined against experimental results, with the 0.717 cABN having the highest predictive accuracy for each set of the investigation.

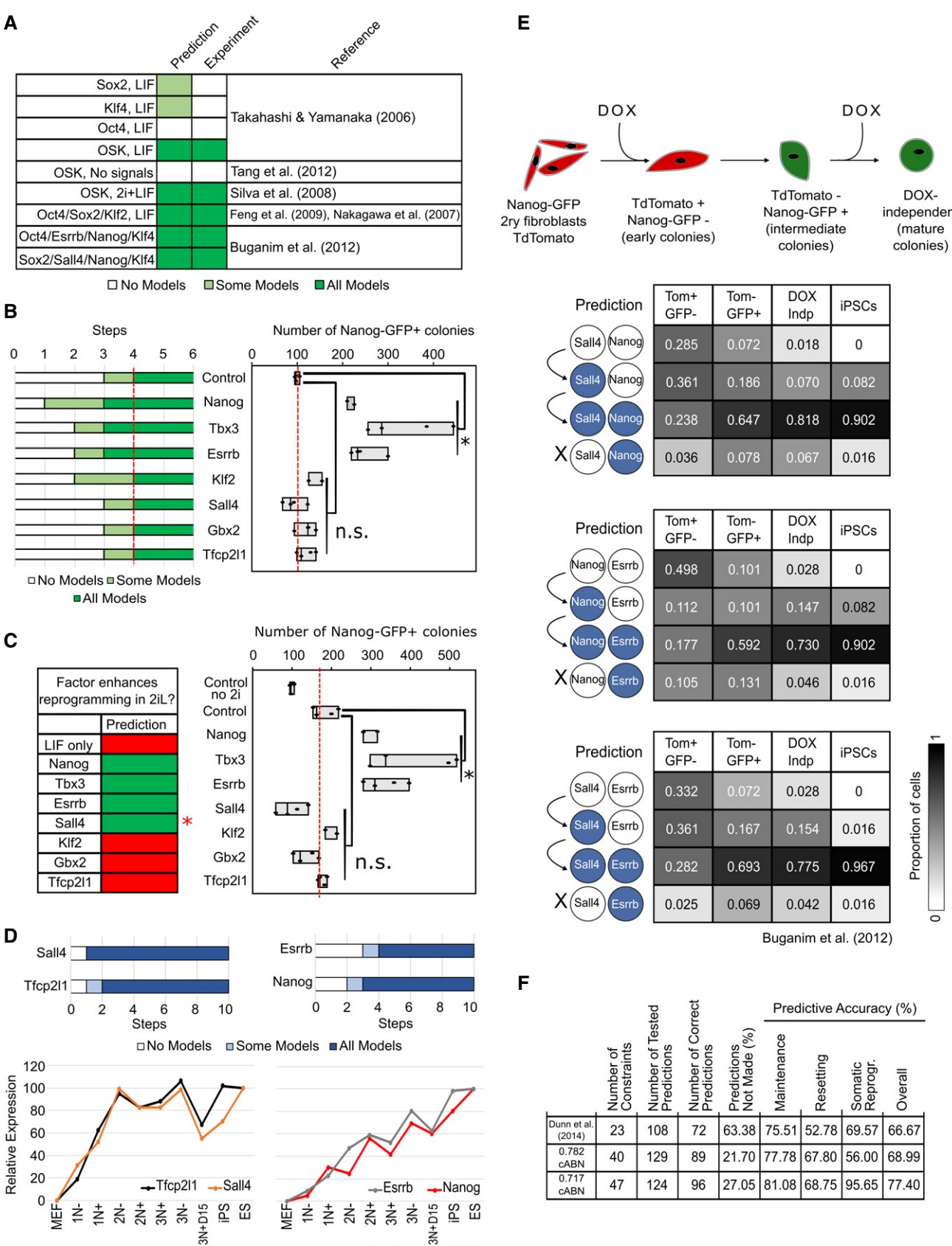

**Figure 7.**

indicate that the 0.717 cABN is consistent with, and predictive of, the majority of behaviours in MEF reprogramming.

The 0.717 cABN could also predict the dynamics of gene activation, again by computing the number of steps required for each component of the network to be stably activated. Compared with gene expression measurements both at population and single-cell level from two independent studies (Buganim *et al*, 2012; O'Malley *et al*, 2013), the 0.717 cABN correctly predicted that Tfcp2l1 and Sall4 activate before Nanog and Esrrb in MEF reprogramming (Fig 7D and E). The predicted sequential activation of gene pairs was validated by comparing the proportion of cells expressing individual genes at different stages of MEF reprogramming (Buganim *et al*, 2012) at single-cell resolution (Fig 7E). Taken together, these analyses suggest that a common gene regulatory program for naïve state maintenance governs reprogramming both from EpiSCs and somatic cells.

To confirm the predictive capacity of our final set of models, and compare with previous iterations—that described by Dunn *et al* (2014) and the 0.782 cABN (Fig 1F, Table EV3)—we used the 0.717 cABN to formulate predictions previously generated for both naïve state maintenance (Dunn *et al*, 2014) and EpiSC resetting (Table EV3). In total, the 0.717 cABN was constrained against 47 experimentally observed behaviours and generated a further 96 predictions consistent with experimental observations. When compared to the previous generations of cABNs, we observed a progressive increase in overall predictive accuracy as we refined the models against new data (Fig 7F).

# Discussion

In this study, we undertook an iterative computational and experimental approach to uncover the logic of resetting post-implantation derived EpiSCs to the ESC state of naïve pluripotency. Our method exploited the power of automated reasoning to constrain a set of possible network models against existing experimental observations, and subsequently to use this set of models to formulate predictions of untested behaviour. The complete set of predictions is summarised in Table EV3. Our results reveal that the biological program ruling maintenance of the naïve state also governs installation of naïve pluripotency both from primed EpiSCs, and from developmentally distant somatic cells. The program that we have progressively refined captures a complex and rich set of behaviours and thereby encapsulates the robust nature of the naïve state captured in 2i+LIF, as well as the fragility of resetting and its dependency of the availability of specific factors. Furthermore, the program is highly predictive: of 124 tested predictions for the 0.717 cABN, 77.4% were supported by experiment. We conclude that maintenance and induction of naïve pluripotency are under the control of the same biological program, which responds dynamically to the initial cell state and signals provided.

Initially, we investigated how forced expression of individual network components influences EpiSC resetting. The program forecast correctly that only some factors—Klf2, Klf4, Esrrb and Tbx3—strongly enhance EpiSC resetting, and furthermore that certain pairs of factors act synergistically. Co-expression of Esrrb with Klf4 or Tbx3 produced a highly efficient resetting context, which permitted us to dissect the sequence of gene activation at the single-cell level.

Significantly, we could identify TFs that can be compensated for by other components during self-renewal, but are stringently required during EpiSC resetting.

Klf2, but not Klf4, was unexpectedly identified as critical for resetting. Yet Klf2 becomes dispensable after the naïve network is established due to functional redundancy with Klf4. We conclude that EpiSC resetting is a conditional process that is highly dependent on the sequence of gene activation, whereas the naïve state maintenance circuitry is robust due to layers of redundancy that confer network resilience (Martello & Smith, 2014).

An often overlooked aspect of modelling is the insight to be gained from analysing incorrect predictions. The distinction between Klf2 and Klf4, which are both members of the Krüppel-like family of TFs and share high sequence homology in the DNA binding domain, was neither predicted nor intuitive. In both naïve state maintenance and somatic cell reprogramming, these genes have been shown to have largely redundant function (Nakagawa *et al*, 2007; Jiang *et al*, 2008; Yeo *et al*, 2014). Furthermore, expression of Klf2 and Klf4 has a similar and potent effect on EpiSC resetting. However, only Klf2 is required for transgene-free EpiSC resetting. This can be understood in the context of the network by examination of the kinetics of gene activation. Klf2 is upregulated within the first 2 h of resetting, whereas Klf4 becomes stably expressed only after 48 h. Thus, inactivation of Klf2 leaves the cell devoid of both Krüppel-like family TFs for the first 2 days. Associated with this, naïve markers normally activated subsequently are not induced and resetting does not proceed. Inactivation of Klf4, in contrast, can be compensated for by the presence of Klf2, which is activated early and maintained throughout. The functional redundancy between Klf2 and Klf4 is exemplified in the observation that *Klf2* KO cells can be reset by transient expression of either Klf2 or Klf4.

Like Klf2, Stat3 is also a factor specifically required for resetting, and the potent effect of the LIF/Stat3 axis in resetting was previously reported (Yang *et al*, 2010; Martello *et al*, 2013; Carbognin *et al*, 2016). Here, we clarified the downstream mediators of Stat3 and observed that three direct targets, Klf4, Tfcp2l1 and Gbx2, cooperatively induce naïve pluripotency. Indeed, only their triple inactivation phenocopies the loss of Stat3 in GOF18 EpiSCs, while single expression of each is sufficient to rescue Stat3 knockdown. We previously found that Tfcp2l1 is required for the resetting of OEC2 EpiSCs, which do not convert spontaneously (Martello *et al*, 2013). In such cells, LIF/Stat3 signalling results in the activation of Tfcp2l1 but not of Klf4 (Martello *et al*, 2013). Moreover, Klf2 induction is attenuated in OEC2 compared to GOF18 EpiSCs (Fig 4 and Appendix Fig S3F). The lack of robust activation of Klf2 and Klf4 may explain the dependency on Tfcp2l1 for OEC2 resetting. Notably, however, other findings, such as Esrrb/Klf4 dual factor synergy and *Klf2* KO phenotype, have been confirmed in OEC2 cells. It is well known that EpiSC lines vary in their properties, including efficiency of resetting (Bernemann *et al*, 2011b; Illich *et al*, 2016). This is consistent with the contingencies revealed by our models.

It is currently debated whether acquisition of naïve pluripotency is an ordered process, following a precise sequence of events, or a stochastic system in which individual cells follow different trajectories. Our results suggest that productive EpiSC resetting is not stochastic, revealing a sequential gene activation trajectory towards the naïve state that is substantiated even at single-cell resolution. This may seem counterintuitive, given that some EpiSCs fail to reset

in the presence of transgenes (Fig 4C) and activation of somatic lineage markers can be detected (Fig 6D). However, we hypothesise that EpiSC resetting is deterministic subject to an initial activation threshold. Technical impedance, such as variable transgene expression and biological stochasticity, may render some cells irresponsive or aberrantly responsive. Crucially however, once cells overcome the initial activation threshold they follow a deterministic trajectory.

Our analyses identified two distinct kinetics of network gene activation (Fig 2B and D). Factors such as Stat3, Tfcp2l1 and Klf2 are rapidly upregulated in 2i+LIF, followed later by factors such as Klf4 and Esrrb. These different gene activation kinetics could be associated with different roles in naïve network installation. Rapidly activating factors are important to initiate naïve network activation, consistent with the observation that Stat3 and Klf2 are essential to resetting (Fig 6A). Slow-activating factors such as Esrrb could play a consolidating role in network installation. In line with this, Esrrb activation is a rate-limiting step for resetting, and Esrrb is one of the most potent factors to induce the naïve state (Fig 6I). This is consistent with the recent finding that Esrrb acts as a pioneering factor in chromatin remodelling for core pluripotency TF recruitment in EpiSC resetting (Adachi *et al*, 2018). We speculate that different modes of activation for genes with different functions could be integral to the information-processing performed by a cell. Understanding how regulation modes are coupled to biological function in a given process may contribute insight into biological computation, and in turn enable the artificial synthesis of molecular logic to achieve a desired cellular behaviour.

Finally, we demonstrated that the network program derived from observations of naïve state maintenance and EpiSC resetting has both explanatory and predictive power in somatic reprogramming. This further suggests that the late phase of somatic reprogramming is deterministic (Buganim *et al*, 2012; O'Malley *et al*, 2013), but also highlights that a common network logic governs acquisition of naïve pluripotency from different starting cell types.

Although arguably the mouse naïve pluripotency network is one of the most well-characterised GRNs, we consider that our methodology could be applied to study other networks with less complete knowledge. Given a preliminary set of network components and interactions, the methodology has the flexibility to incorporate or eliminate constraints and regulators. Importantly, it can evaluate network behaviour against experimental observations and guide network refinement towards higher predictive accuracy and reality.

Furthermore, our approach is complementary to computational modelling approaches that typically consider a single network and explore its dynamics under asynchronous updates (Xu *et al*, 2014; Abou-Jaoudé *et al*, 2016; Yachie-Kinoshita et al, 2018). It is a significant challenge to select the right model to investigate given uncertainty in the set of interactions, and it is difficult to reason over multiple experiments in the process of model formulation. We provide an automated platform to enrich for models that are provably consistent with multiple biological observations. From this set, the software can readily identify the "minimal model", which has the fewest interactions. Indeed, it is a common strategy to develop the most parsimonious interaction network (Muñoz Descalzo *et al*, 2013; Xu *et al*, 2014). We used the minimal model from the 0.717 cABN to investigate the dynamics of resetting simulated under asynchronous updates (Appendix Fig S8B–F and Appendix Text S1), demonstrating a predictive accuracy of 75.86%. This highlights that

our method can be exploited to identify a candidate network topology as a starting point for subsequent investigation using alternative methodologies. However, the increasing wealth of observed heterogeneity in cellular gene expression and behaviours could indicate that multiple network topologies co-exist in cell state regulation. Given our approach can capture highly predictive sets of models with consistent behaviours, we provide an ideal starting point to unpack biological heterogeneity at the network level.

The cABN implicitly defines different concrete networks that are consistent with the imposed constraints, but may not generate the same dynamic behaviour in response to a tested perturbation. When all concrete networks generate the same response, we formulate a definitive prediction that is subsequently tested experimentally. We then calculate the fraction of correct predictions, which reveals the predictive accuracy of the cABN. In those cases where not all concrete networks agree, we are unable to form a prediction without prioritising one network over another. However, these cases of "no prediction" in fact reveal discriminating experiments that can be performed to constrain the cABN further. Interestingly, during the process of cABN refinement (Fig 1A) we observed a significant reduction in the percentage of "no prediction" with a concomitant increase in the accuracy of the predictions made (Fig 7F and Table EV3). It could be argued that the absence of a prediction should be considered as an incorrect prediction, because the cABN could not definitively confirm the correct behaviour. However, even in this extreme case the predictive accuracy of the final 0.717 cABN is 65.1%, therefore remaining highly predictive.

In summary, our analyses point to a common biological program that governs naïve pluripotency maintenance and induction. The power and utility of the combined computational and experimental methodology is exemplified by predicting the sequence of gene activation that occurs during EpiSC and somatic reprogramming, even at single-cell resolution, and pinpointing which factors affect resetting efficiency. This method enabled the identification of pairs of TFs that dramatically accelerate resetting, yielding an overall efficiency increase of up to 50-fold. The refined cABN provides a platform for revealing principles of network dynamics underlying pluripotency transitions, including the emergence and dissolution of naïve pluripotency in the embryo (Boroviak *et al*, 2015). Moreover, a similar iterative methodology using the RE:IN tool (Yordanov *et al*, 2016) could be applied to study direct lineage reprogramming (Davis *et al*, 1987; Xie *et al*, 2004; Vierbuchen *et al*, 2010). We further envisage that our approach should be applicable to derive an understanding of network architecture and dynamics underpinning other cell fate transitions.

# Materials and Methods

### Cell lines

All EpiSC lines in this work were cultured as described in Guo *et al* (2009) on fibronectin-coated plates in serum-free media N2B27 (DMEM/F12 and Neurobasal [both Life Technologies] in 1:1 ratio, with 0.1 mM 2-mercaptoethanol, 2 mM L-glutamine, 1:200 N2 [Life Technologies] and 1:100 B27 [Life Technologies]) supplemented with FGF2 (12 ng/ml) and Activin (20 ng/ml) produced in house. GOF18 EpiSCs were described in Han *et al* (2010a) and generously

provided by Hans Schöler. OEC2-Y118F (Oct4-GFP) EpiSCs were described in Yang *et al* (2010). TNGA MEFs were cultured as described in O'Malley *et al* (2013).

## Plasmid constructions

Individual core pluripotency network factors were either amplified from cDNA or cloned from existing expression plasmids into pENTR2B donor vector. Subsequently, the transgenes were gateway cloned into the same destination vector containing PB-CAG-DEST-bghpA and pGK-Hygro selection cassette. The sizes of final expression constructs ranged from 8.5 to 10.7 kb.

To construct the T2A linked inducible overexpression plasmids, Esrrb and either Tbx3 or Klf4 were PCR amplified with part of the T2A sequence flanking the 3′ or 5′ of the gene, respectively. Three-way ligation of both gene fragments together with pENTR2B vector resulted in the fusion of EsrrbT2ATbx3 or EsrrbT2AKlf4. Subsequently, the fusion constructs were gateway cloned into the same final destination vector containing TRE-CMV and a Venus reporter. To generate co-expression cell lines, cells were co-transfected with a plasmid containing rtTA and a Neomycin selection cassette.

## Transient overexpression of factors for EpiSC resetting

1.5 μg of plasmid DNA was transfected with 3 μl of Lipofectamine 2000 to $1 \times 10^5$ EpiSCs in suspension in FGF2/ActivinA containing N2B27 medium with Rock inhibitor Y-27632 (Sigma, 1:1,000) overnight in one well of the 12-well plate. The next day, medium was switched to 2i+LIF medium to initiate reprogramming. GFP-positive colonies were scored at Day 7 of reprogramming. When a combination of two factors was co-transfected, 0.75 μg of plasmid DNA of each factor was used. For the control single factor transfections, 0.75 μg of plasmid DNA harbouring the indicated factor together with 0.75 μg of empty vector plasmid was used.

## Generation of KO EpiSCs with CRISPR/Cas9

The gRNA pair was chosen to delete the largest coding exons within Klf2 and Klf4 to ensure complete loss of function. The gRNA design was conducted using online CRISPR gRNA design tool https://www.dna20.com/eCommerce/cas9/input. The chosen gRNAs were based on the minimal off-target scores. The gRNA containing plasmids were cloned by annealing the complementary oligos indicated in Appendix Table S3, and cloned into BbsI digested pX458 vector (Addgene). The constructs were sequence validated before transfection.

A pair of gRNA containing plasmids based on px458 designed with specific deletion were transfected using Lipofectamine 2000 (Invitrogen). 500 ng of each plasmid was transfected with 3 μl Lipofectamine 2000 to $2 \times 10^5$ EpiSCs in suspension in Activin/FGF2/XAV939 containing N2B27 medium with Rock inhibitor Y-27632 (Sigma, 1:1,000) overnight in one well of the 12-well plate. The next day, the media was refreshed with Activin/FGF2/XAV939/Rock inhibitor, and 48 h post-transfection, 2,000 GFP high cells were sorted into 6 cells for colony formation. Individual colonies were picked and genotyping was conducted from extracted genomic DNA by primers indicated in Appendix Table S3 to identify clones with designed deletion. For *Klf2* KO, deletion from both gRNAs resulted

in genotyping PCR product to shift from 890 bp representing the wild-type allele to 130 bp. For *Klf4* KO, deletion from both gRNAs resulted in genotyping PCR product to shift from 840 bp representing the wild-type allele to 100 bp. Only homozygous mutants were chosen for subsequent experiments.

## siRNA knockdown for EpiSC resetting

Final concentration of 20 nM siRNAs together with 0.5 μl of Dharmafect 1 (Dharmacon, T-2001-01) was transfected to $1 \times 10^5$ EpiSCs in suspension in Activin/FGF2 containing N2B27 medium with Rock inhibitor Y-27632 (Sigma, 1:1,000) overnight in one well of the 12-well plate. At least 2 siRNAs were used for each target gene knockdown, and the catalogue numbers of all siRNAs are shown in Appendix Table S1. The next day, medium was switched to 2i+LIF to initiate reprogramming. GFP-positive colonies were scored at Day 7 of reprogramming.

## siRNA knockdown for ESC maintenance

To test the effect of knockdown of individual factors on maintenance of naïve pluripotency, we transfected siRNA in mES cells and replated them after 48 h at clonal density, as described in Dunn *et al* (2014). Five days after plating, we scored the number of pluripotent colonies, relative to cells transfected with a control siRNA. At least 2 siRNAs were used for each target gene knockdown, and the catalogue numbers of all siRNAs are shown in Appendix Table S1.

## EpiSC resetting of DOX-inducible factor combinations

Cells with the stably integrated *piggyBac* transposase (500 ng), *piggyBac* transposon constructs harbouring the DOX-inducible factor combinations (375 ng) and rTtA construct (125 ng) were plated in N2B27 medium containing F/A. The next day, medium was switched to 2i+LIF with or without DOX 0.2 μg/ml for 2 days to induce transgene expression. At Day 2, medium was switched to 2i + LIF without DOX. Images were acquired at Day 6, and clonal assays were performed at Day 2-4-6-8 (See also Appendix Fig S3D). For single-cell qPCR analysis of resetting intermediates, cells were kept in 2i+LIF+DOX throughout the experiment for up to 4 days. Clonal assay was performed by replating 250 cells in one well of a 12-well plate in 2i+LIF without DOX.

## MEF reprogramming

All MEF reprogramming experiments were conducted on primary MEFs at P2. $2.5 \times 10^5$ TNGA MEFs were transfected with 2.5 μg of OSKM *piggyBac* transposon construct (Yusa *et al*, 2009), 2.5 μg of naïve factor *piggyBac* transposon construct or empty vector, together with 1.9 μg of HyPBase (Yusa *et al*, 2011) using NEON transfection system (Thermofisher). Transfected cells were plated in MEF medium, and the next day, one-tenth of cells were replated into 1 well of a 6 well with MEF medium supplemented with LIF, 50 μg/ml ascorbic acid and 500 nM Alk inhibitor A83-01, as described in O'Malley *et al* (2013). The Nanog-GFP⁺ colonies were scored at Day 12. For experiments with 2i addition, 2i was added to MEF reprogramming media from Day 6 onwards.

## Alkaline phosphatase staining

For AP staining, cells were fixed with a citrate–acetone–formaldehyde solution and stained using the Alkaline Phosphatase kit (Sigma, cat. 86R-1KT). Plates were scanned using a Nikon Scanner and scored manually.

## RNA extraction, reverse transcription and Real-time PCR

Total RNA was isolated using RNeasy kit (Qiagen), and DNase treatment was conducted either after RNA purification or during column purification. cDNA was transcribed from 0.5 to 1 μg RNA using SuperScriptIII (Invitrogen) and oligo-dT priming. Real-time PCR was performed using on StepOnePlus and QuantStudio machines (Applied Biosystems). Target gene primer sequences and probes used are listed in Appendix Table S2. Expression levels were normalised to Actinβ or Gapdh. Technical replicates for at least two independent experiments were conducted. The results were shown as mean and standard deviation calculated by StepOnePlus software (Applied Biosystems).

## Single-cell gene expression profiling

OpenArrays were custom designed by ThermoFisher with the Taqman assay ID shown in Appendix Table S4. Single cells were directly deposited by Fluorescence Activated Cell Sorting into 9 μl of a pre-amplification mixture (CellDirect One-Step qRT-PCR kit, 11753-500) which contains 0.05× of each TaqMan assay, 1× Cell-Direct reaction mix, 200 ng/μl SuperscriptIII/Platinum Taq and 100 ng/μl SUPERase-In (ThermoFisher) in DNA suspension buffer (TEKnova). The reverse transcription and gene-specific PCR amplification was carried out in a thermal cycler with the following condition: 50°C for 30 min, 95°C for 2 min followed by 24 cycles of 95°C for 15 s, 60°C for 4 min. cDNA was diluted 1:10, and only cells with at least two housekeeping genes amplified were chosen for whole panel gene expression profiling. The cDNA samples were loaded onto an OpenArray using OpenArray AccuFill system, and the quantitative real-time PCR was run using Quantstudio 12K Flex System. For gene expression analysis, the average of five housekeeping genes (Actβ, Gapdh, Tbp, Ppia, Atp5a1) was used for normalisation.

For quality control, the expression of Actβ and Atp5a1 was first analysed to establish whether the cells were deposited successfully. We excluded wells where no amplification or abnormal amplification was obtained.

## RNA sequencing

RGd2 mouse ESCs were derived and expanded in 2i for six passages and subsequently cultured in defined conditions on gelatin-coated plates for five passages in N2B27 basal medium supplemented with four combinations of cytokine LIF (20 ng/ml), GSK3 inhibitor CHIR99021 (CH, 3 μM) and MEK inhibitor PD0325901 (PD, 1 μM): PD+CH, PD+LIF, CH+LIF and PD+CH+LIF. The cells were passaged every 3 days at a density of 15,000 cells per cm$^2$ with medium refreshed daily.

Total RNA was isolated with RNeasy RNA purification. Ribozero rRNA-depleted RNA was used to generate sequencing libraries. Single-end sequencing was performed, and the reads were mapped using NCBI38/mm10 with Ensembl version 75 annotations. RNA-seq reads were aligned to the reference genome using tophat2. Only uniquely mapped reads were used for further analysis. Gene counts from SAM files were obtained using htseq-count with mode intersection non-empty, -s reverse. Differential gene expression analysis was conducted using Bioconductor R package DESeq2 version 1.4.5. DESeq2 provides two *P*-values, a raw *P*-value and a Benjamini–Hochberg *P*-value (adjusted *P*-value). An adjusted *P*-value threshold of 0.05 was used to determine differential gene expression (95% of the results are not false discoveries, error rate 0.05 = 5%). The data are available at the NCBI Gene Expression Omnibus (accession number: GSE111694).

## Chimera production

Adult chimeras were produced by microinjection of iPSCs into C57BL/6 blastocysts. In total, we used 10–15 animals, including collecting superovulated embryos and pseudopreganant hosts for embryo transfer. Animal studies were authorised by a UK Home Office Project License and carried out in a Home-Office-designated facility.

## Identifying possible interactions

The initial 0.832 ABN (Appendix Fig S1C) was constructed from a set of *definite* interactions downstream of LIF, CH and PD, based on previous experimental studies that identified the direct targets of these signals (Niwa *et al*, 2009; Silva *et al*, 2009; Martello *et al*, 2012, 2013; Qiu *et al*, 2015), and a set of *possible* interactions derived from our RNA-Seq and RT–qPCR datasets as follows. Alternative network models are thereby implicitly defined by the ABN, as the unique instantiation of $n$ possible interactions defines $2^n$ concrete networks.

Seven Pearson coefficients were generated for each gene pair, one from each of seven datasets, which quantify the correlation in gene expression under the action of different combinations of LIF, CH and PD. An interaction between two genes was defined to be possible and positive if at least one of these coefficients was above a given threshold, and the majority of the remaining coefficients were greater than zero. Similarly, an interaction between two genes was defined to be possible and negative if at least one of these coefficients was below the negative of a given threshold, and the majority of the remaining coefficients were less than zero. In cases where there were positive coefficients above the threshold as well as negative coefficients below the threshold, we let the majority rule. Given that correlations alone do not reveal which gene behaves as the regulator, possible interactions were defined to be bidirectional.

We identified the Pearson correlation threshold by constructing a set of experimental constraints (Fig 1C and Appendix Fig S1F, as described below). We then sought the maximum Pearson coefficient threshold that generated a set of possible interactions that could satisfy these expected behaviours, using the RE:IN software to test for satisfiability. In doing so, we minimised the number of possible interactions and therefore the number of concrete models in the ABN.

## Discretising gene expression measurements and encoding experimental observations

We discretised the gene expression profile of GOF18 EpiSCs (Appendix Fig S1E) by setting a gene to High if its expression was at least 0.5 of its level in mouse ESCs in 2i+LIF. We therefore discretised the GOF18 EpiSC state to be such that Oct4, Sox2 and Sall4 were High, while the remaining TFs were Low. MEK/ERK and Tcf3 were also set to High in these cells, as they are cultured in F/A.

We added a set of experimental observations to our existing set of constraints concerning maintenance of naïve pluripotency (Dunn *et al*, 2014), by discretising gene expression profiles for the following experimental behaviours, shown in Appendix Fig S1F and summarised in Table EV2:

**Control**: If none of the pluripotency factors are initially expressed, then 2i + LIF alone is insufficient to reach the naïve state, which is defined to be the gene expression state of mouse ESCs cultured in 2i + LIF(Dunn *et al*, 2014).

**EpiSC in 2i + LIF**: Starting from the discretised gene expression profile of GOF18 EpiSCs, 2i + LIF is sufficient for these cells to reset and stabilise in the naïve state (Han *et al*, 2010a; Martello *et al*, 2013).

**EpiSC in 2i only**: 2i alone is insufficient to reset GOF18 EpiSCs (Martello *et al*, 2013).

**EpiSC in 2i with Tfcp2l1 expression**: Forced expression of Tfcp2l1 is sufficient to reset GOF18 EpiSCs in 2i alone (Martello *et al*, 2013).

**Nanog knockout EpiSCs in 2i + LIF**: Nanog knockout prevents EpiSCs from reaching the naïve state in 2i + LIF (Stuart *et al*, 2014).

**Nanog knockout EpiSCs in LIF + CH**: Nanog knockout EpiSCs in the presence of LIF + CH is sufficient to activate Oct4, Esrrb, Klf2, Tfcp2l1, Klf4 and Stat3 (Stuart *et al*, 2014).

Each constraint consists of an initial and final discrete gene expression pattern, which are the required initial and final states of network trajectories that correspond to the experiment under consideration. We allow 20 steps for each experiment trajectory to stabilise. The final state is either unreachable (indicated by a bar over the final time step in Appendix Fig S1F), or stable (indicated by an asterisk). In the case where the full gene expression state cannot be defined (e.g. Tfcp2l1 forced expression in 2i), then we define the final state at two sequential steps to ensure that the key genes are sustained.

We encoded these constraints together with the ABN into the RE:IN tool (Yordanov *et al*, 2016). RE:IN synthesises only those concrete network models consistent with this set of expected behaviours, that is, which generate trajectories that pass through the required expression states. These concrete networks comprise the cABN.

When investigating the gene activation kinetics of resetting in the 0.782 cABN, we included the observation that forced expression of Sall4 in GOF18 EpiSCs does not increase the efficiency of resetting to the naïve state (Fig 1E). To ensure that this holds for all concrete models of the cABN, we encoded a new constraint that defined when Sall4 expression is imposed, an EpiSC will not reach the naïve state at an earlier step than the case in which it is not, regardless of the step at which the latter occurs. This is illustrated in Table EV2.

We explored the sensitivity of our approach to missing components by testing whether the above constraints are satisfiable if each component is removed individually. For all components save Esrrb, we found removing the component from the ABN prevents the constraints from being satisfied. This demonstrates that these components are absolutely required to generate the expected behaviour of ESCs and EpiSC reprogramming. Removal of Esrrb along with the 5 constraints concerning Esrrb knockdown or forced expression yields a cABN that can satisfy the remaining constraints, but cannot explain known Esrrb phenotypes and has low predictive power.

### Network dynamics

Each concrete network model in the ABN is considered as a state transition system, with a deterministic update scheme. Dynamic behaviour emerges from the update functions that are applied to each component, which are logical functions that define how the gene updates its state in response to its regulators. Often such update functions are defined according to the named regulators of a given target, but for an ABN, the regulators of a target can vary between concrete models. We therefore defined a set of twenty update functions that are not dependent on named regulators (Yordanov *et al*, 2016), which reason about whether some, all or none of a targets activators/repressors are present. In this present study, we consider a subset of these conditions (regulation conditions 0–8 as described in Yordanov *et al* (2016)), which assume that a gene requires at least one activator in order to be expressed. In a concrete model, each component is assigned one of these regulation conditions to ensure that the constraints are met.

### Required and disallowed interactions

We characterise the cABN by identifying which of the possible interactions are common to all concrete networks—required interactions—and which are never present—disallowed interactions. A simple algorithm is implemented that first identifies a single concrete network consistent with the experimental observations. Each of the possible interactions that are instantiated as present in this example solution is subsequently removed individually from the ABN, and RE:IN identifies whether the constraints are satisfiable in the absence of the interaction. If the constraints are unsatisfiable when a given possible interaction is removed, then it must be the case that it is present in every concrete network that satisfies the constraints. Conversely, we examine all interactions not present in the example solution that we initially found, testing whether the constraints are still satisfiable if these interactions are individually imposed as definite. If once a possible interaction is switched to being definite and the constraints are no longer satisfiable, we conclude that this particular interaction can never be present in any concrete model solution. The remaining interactions—those which can be removed or imposed individually without preventing the constraints from being satisfied—remain as possible and will be needed in some concrete models, but not all.

### Formulating model predictions

Via RE:IN, our approach automatically synthesises concrete network models consistent with the expected behaviour of the experimental system. To formulate predictions of untested behaviour, we interrogate the entire set of consistent models. A prediction is generated, and tested, only when all models are in agreement. The behaviour

of only a subset of models, which may not be fully representative, is never tested experimentally. However, such cases reveal discriminating experiments that, if performed, would allow us to eliminate a subset of candidate models from the cABN.

To generate predictions, hypotheses are encoded as additional constraints, and we test whether they are satisfiable together with the set of experimental constraints. Crucially, we also test the null hypothesis—that under the same conditions the expected behaviour cannot be obtained. If both are satisfiable independently, then it must be the case that some models satisfy the hypothesis, while others satisfy the null hypothesis. If all models satisfy the hypothesis, while the null is unsatisfiable, then a prediction that the hypothesis holds can be formulated. If the null hypothesis is satisfiable, while the hypothesis is unsatisfiable, then a prediction can also be made, which is that the expected behaviour is never observed.

For example, to test whether GOF18 EpiSCs can reset to the naïve state under Gbx2 knockdown in 2i+LIF, we formulate a constraint with these initial and final states. We then also formulate and test the constraint that GOF18 EpiSCs do not reach the naïve state under Gbx2 knockdown in 2i+LIF. In this particular example, we found that our hypothesis constraint was satisfiable, while the null hypothesis constraint was unsatisfiable. Therefore, all concrete models predict that GOF18 EpiSCs will reset in 2i+LIF with Gbx2 knockdown, which was subsequently found to be consistent with experimental evidence.

### Identifying the number of regulation steps to reach the naïve state

To determine how many regulation steps are required to stabilise in the naïve state, starting from the EpiSC state, we formulate hypotheses for each possible case, e.g. that it stabilises at Step 2, at Step 3, etc. As described above, for each case we also test the null hypothesis. In this manner, we deduced whether some, all or none of the models allowed EpiSCs to stabilise in the naïve state at a given regulation step.

RE:IN allows the user to implement an asynchronous scheme, in which a single gene updates at each step, and the order in which genes update is chosen non-deterministically. Under this scheme, if RE:IN determines that the constraints are satisfiable, this only ensures that there exists at least one model and path that is consistent with each constraint. That is, it is possible that the genes could update in a different order and reach a different state from the same initial conditions. Formulating predictions for the number of steps for all models to stabilise in the naïve state under an asynchronous update scheme would require further assumptions to be made. Either a limit would have to be placed on the maximum number of sequential updates for a specific gene, or a restriction to ensure that all genes update within a certain number of steps. It would also be important to consider what is considered "fair" in implementing asynchronous updates, to avoid unrealistic scenarios such as the same gene repeatedly updating and no others.

### K-means clustering to discretise single-cell gene expression data

We used $k$-means clustering with $k = 2$ on the log10-transformed single-cell gene expression data (Fig 4D) in order to discretise gene expression. We identified a unique discretisation threshold for each gene. Of note, the mean expression levels in the two clusters differed by several orders of magnitude.

### SPADE analysis

We conducted a SPADE analysis using SPADEV3.0 (Anchang *et al*, 2016), using the default settings. This was carried out on the log10-transformed single-cell gene expression data (Fig 4D).

### Asynchronous simulations

We carried out a comparison of the predictions generated by the 0.717 cABN, which assumes synchronous updates, with the minimal model (Appendix Fig S8A) simulated under an asynchronous scheme. Simulations were run using BooleanNet (Albert *et al*, 2008). Full details are provided in Appendix Text S1, and the results are presented in Appendix Fig S8B–F.

### Statistical analysis

We used the two-tailed unpaired Student's *t*-test with $P < 0.05$ to define statistical significance, unless specified otherwise. The number of the experiments ($n$), the dispersion and precision measurements (mean, median, standard errors and standard deviations) can be found in figure legends.

## Data availability

The RNA-sequencing data are available at the NCBI Gene Expression Omnibus (accession number: GSE111694). The files used to generate the cABN are available at research.microsoft.com/rein, which also provides a tutorial for the tool, and FAQ.

*Expanded View* for this article is available online.

### Acknowledgements

We thank members of the Smith and Martello laboratories, and Boyan Yordanov and Christoph Wintersteiger from Microsoft Research for advice and discussion. We are grateful to Jose Silva, Sirio Dupont, Michelangelo Cordenonsi and Marco Montanger for critical reading of the manuscript. We thank Ge Guo for providing naïve factor constructs, Kosuke Yusa for OSKM reprogramming *piggyBac* transposon and HyPBase transposase constructs, Jose Silva for TNGA MEFs. We also thank Yosef Buganim for sharing the MEF reprogramming single-cell RT–qPCR data. Sarah Gharbi and Anzy Miller for their assistance with the OpenArray system. We thank Andy Riddell for assistance for FACs sorting. S-J.D. is supported by Microsoft Research. G.M.'s laboratory is supported by grants from Giovanni Armenise-Harvard Foundation and Telethon Foundation (TCP13013). A.S. and M.A.L. are funded by the RCUK | Biotechnology and Biological Sciences Research Council (BBSRC). The Cambridge Stem Cell Institute receives core funding from the Wellcome Trust and Medical Research Council. M.A.L. was a Sir Henry Wellcome Postdoctoral fellow and received support from the University of Cambridge Institutional Strategic Support Fund. AS is a Medical Research Council professor.

### Author contributions

S-JD carried out the computational modelling. MAL, EC and GM carried out the experiments. S-JD, MAL and GM analysed computational predictions and

experimental data. S-JD, MAL, GM and AS designed the study and wrote the paper. AS and GM supervised the study.

## Conflict of interest

Sara-Jane Dunn is a Scientist at Microsoft Research (MSR), which is part of the Microsoft organisation. MSR carries out fundamental research in Computer Science and related disciplines. All other authors declare that they have no conflict of interest.

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
