## [Review Process File · The EMBO Journal]

A common molecular logic determines embryonic stem cell self-renewal and reprogramming

Sara-Jane Dunn, Meng Amy Li, Elena Carbognin, Austin Smith and Graziano Martello

Review timeline:

Submission date:	7th Jun 2018
Editorial Decision:	20th Jul 2018
Revision received:	7th Aug 2018
Editorial Decision:	27th Aug 2018
Revision received:	25th Sep 2018
Editorial Decision:	2nd Oct 2018
Revision received:	3rd Oct 2018
Accepted:	4th Oct 2018

Editor: Daniel Klimmeck

Transaction Report:

(Note: Please note that the manuscript was previously reviewed at another journal and the reports were taken into account in the decision making process at The EMBO Journal. Since the original reviews are not subject to EMBO's transparent review process policy, the reports and author response cannot be published here. With the exception of the correction of typographical or spelling errors that could be a source of ambiguity, letters and reports are not edited. The original formatting of letters and referee reports may not be reflected in this compilation.)

1st Editorial Decision

20th Jul 2018

Thank you again for the submission of your manuscript (EMBOJ-2018-100003) for consideration at The EMBO Journal. As mentioned, I have evaluated your revised manuscript as well as the referees' reports from the earlier assessment at a different venue together with your rebuttal letters, and in addition we have asked an arbitrating advisor to consider your work and related exchange, whose comments are enclosed below. Thank you also for providing additional information towards the advisor's arguments, which are helpful.

I judge the comments of the advisor to be generally reasonable and can - based on the overall interest and novelty of your findings as well as your sensible preliminary response - offer to invite you to revise your manuscript to address the advisor's concerns.

Please note however, that since - aside of the discussed matters of data representation - the remaining issues are touching highly technical and yet key issues (predictive accuracy on predictions versus constraints, training- versus independent validation data-sets; priorities of different model versions; generality of findings; contextualization of alternative asynchronous modelling strategies) your revised manuscript will need to go back to the advisor specifically to judge these issues. As such, in this case we cannot commit beyond stating we will return the revision and arguments to the modelling expert for one more round and will have to take his-her advice into consideration in arriving at a final decision.

ADVISOR'S COMMENTS:

I share the concerns of reviewer 3 regarding the modelling approach and have further concerns regarding its appropriateness and technical soundness. As already pointed out by reviewer 3, the methodology itself is not novel.

1. The modelling methodology relies on automated formal reasoning which effectively finds a collection of models that satisfy some constraints. However the goal should be to find the real network used by the cell and not a collection of plausible models. Indeed, there is no experimental data suggesting that individual equivalent cells in the same conditions would use different networks to achieve the same end goal. The authors themselves argue for determinism in the biological process they study. The authors offer no workaround to select what would be the preferred model and this brings into question the veracity of the presented models (and their ultimate relevance for the biological process as it is a biology paper and not a computer science paper).

2. There is an absence of rigorous objective criteria to test why one model (or to be more accurate a set of models as the authors themselves point out) is preferred.

For example, the model of Dunn et al. (2014) generally outperforms the .782 and .717 cABN models regarding prediction accuracy (Table S5 which is not commented in the main text) yet all models differ by their set of definite interactions. According to the authors own logic, the Dunn et al. model from 2014 should still stand as the preferred model for ES self-renewal and therefore contradicts the claim of the paper's title.

Authors have a "no prediction" category in their tables. If it means that the model cannot make a prediction, this is a severe drawback as it decreases the predictive power (the authors currently excluded this category in the analysis).

3. Data over-fitting leading to choice of models with opposite behaviour. For example, Klf2 has late activation in Figure 1 and 2 using .832 and .782 cABN and early activation in Figure 6 using .717 cABN during resetting to the ES state. According to the experimental data, Klf2 expression behaves like Stat3 expression (very early activation). However none of the models predicts that. Inconsistent predictions therefore highlight the sensitivity of the models to noise in the training set and the subsequent derivation of models that could be constrained by idiosyncrasies. A sensitivity analysis to the training set is absolutely necessary.

4. Finally reviewer 3 makes a very valid point about the interesting possibilities of asynchronous updating that is dismissed by the authors on unrigorous grounds and should definitely be explored.

We would like to thank the arbitrating reviewer for considering our manuscript and response to previous review. The comments provided have highlighted areas of the manuscript that lack clarity, as well as additional analysis to extend our results. In addition to the specific points raised, we have made the following changes to increase clarity:

- Provided an identical layout for all model visualisation diagrams to avoid confusion between different cABNs.
- Updated the schematic explanation of modelling approach in Fig. 1a.
- Emphasised the separation of training and test data. Our constraints are always kept distinct from predictions and the predictive accuracy we report concerns only predictions, and not constraints. We have further highlighted this point in the text and figures (e.g. Fig. 1a) to avoid any ambiguity.

Point by point response:

I share the concerns of reviewer 3 regarding the modeling approach and have further concerns regarding its appropriateness and technical soundness. As already pointed out by reviewer 3, the methodology itself is not novel.

1. The modeling methodology relies on automated formal reasoning which effectively finds a collection of models that satisfy some constraints. However the goal should be to find the real network used by the cell and not a collection of plausible models. Indeed, there is no experimental data suggesting that individual equivalent cells in the same conditions would use different networks to achieve the same end goal. The authors themselves argue for determinism in the biological process they study. The authors offer no workaround to select what would be the preferred model and this brings into question the veracity of the presented models (and their ultimate relevance for the biological process as it is a biology paper and not a computer science paper).

We agree that the primary goal of modelling biological systems is to provide insight into the 'real' network implemented by cells. Our approach exposes the challenge of identifying the single correct network, when many alternative models are consistent with the available experimental evidence. While investigations based on a single model rely on the 'right' model being chosen at the outset, we provide a route to consider a set of potential models until invalid models can be ruled out by additional observations.

It is correct that no data show definitively that similar cells use different regulatory networks, but conversely, one could also argue that a single network topology may be inconsistent with the wealth of experimental observations of heterogeneity in cellular gene expression and behaviours, such as cellular reprogramming. Our conclusion regarding determinism is not contradictory, but highlights that there is order in the system despite the heterogeneity, in an analogous way to asynchronous dynamics.

We accept that a process for uncovering a preferred model was not clearly discussed in the manuscript. Many modelling approaches, such as that by Yachie-Kinoshita et al. (2017), consider the 'simplest' network to be the preferred model. To that end, our approach allows the minimal network to be identified directly. This is the network that has the fewest interactions (Fig. S8a). We have added the following discussion to the manuscript to highlight this issue:

"Furthermore, our approach is complementary to computational modelling approaches that typically consider a single network and explore its dynamics under asynchronous updates^{14,64}. It is a significant challenge to select the right model to investigate given uncertainty in the set of interactions, and it is difficult to reason over multiple experiments in the process of model formulation. We provide an automated platform to enrich for models that are provably consistent with multiple biological observations. From this set, the software can readily identify the 'minimal model', which has the fewest interactions." P28, line 2.

2. There is an absence of rigorous objective criteria to test why one model (or to be more accurate a set of models as the authors themselves point out) is preferred. For example, the model of Dunn et al. (2014) generally outperforms the .782 and .717 cABN models regarding prediction accuracy (Table S5 which is not commented in the main text) yet all models differ by their set of definite interactions. According to the authors own logic, the Dunn et al. model from 2014 should still stand as the preferred model for ES self-renewal and therefore contradicts the claim of the paper's title.

We assume that the reviewer is referring to Table S3, not S5, which lists the siRNAs used in this study. References to Table S3 are provided in the main text on page 23, and also in the Discussion. Please note that the table has been updated in this revision, as we noticed a counting error in the pairwise comparisons. In addition, we have added the predictive accuracy of the Dunn et al. (2014) model for resetting behaviours to help clarify the performance of each cABN.

We elected to use predictive accuracy to measure and compare the performance of the different models. The 0.717 cABN has an overall predictive accuracy of 77.6% compared with 61.24% for the 0.782 cABN and 66.67% for the Dunn et al. (2014) cABN. Furthermore, if we consider each set of tests individually (maintenance single/double factor knockdown, resetting from EpiSCs, resetting from somatic cells) the 0.717 cABN consistently outperforms the previous model iterations in each category. Lastly, related to the comment below, if we were to count 'no prediction' as incorrect, then these figures change to 65.1% for the 0.717 cABN, 57.1% for the 0.782 cABN, and 43.37% for the 2014 cABN. We would therefore argue that the .717 cABN generally outperforms these other models.

Authors have a "no prediction" category in their tables. If it means that the model cannot make a prediction, this is a severe drawback as it decreases the predictive power (the authors currently excluded this category in the analysis).

The reviewer correctly points out that we determine predictive accuracy based only on the number of predictions that are formulated by each cABN. As shown above, even including 'no prediction' as incorrect, the 0.717 cABN has the highest predictive accuracy overall.

Cases of 'no prediction' arise when not all models within the constrained set satisfy the proposed hypothesis. While it could be argued that this is a drawback, such scenarios reveal discriminating

experiments that can be performed to inform the modelling further, which would enable us to eliminate the subset of models that do not satisfy the hypothesis, and thus enrich for the “preferred model”. We would also like to point out that we made and tested a far greater number of predictions than typically presented in model analysis.

3. Data over-fitting leading to choice of models with opposite behavior. For example, Klf2 has late activation in Figure 1 and 2 using .832 and .782 cABN and early activation in Figure 6 using .717 cABN during resetting to the ES state. According to the experimental data, Klf2 expression behaves like Stat3 expression (very early activation). However none of the models predicts that. Inconsistent predictions therefore highlight the sensitivity of the models to noise in the training set and the subsequent derivation of models that could be constrained by idiosyncrasies. A sensitivity analysis to the training set is absolutely necessary.

We appreciate the concern regarding over-fitting. We think that this is borne largely out of a lack of clarity in our figures, with seemingly opposite behaviours of Klf2 in Figs. 1, 2 and 6. In fact Fig. 1d was not derived from any model or experimental data but was included as a schematic to illustrate how individual genes can be activated en route to the naïve state, and how we count the number of steps until the naïve state is reached. We have removed this panel to avoid possible confusion for the reader.

The reviewer is correct that our models failed to predict early activation of Klf2, though they do correctly predict the order of activation relative to other, late-activated genes. We have now added emphasis to the legend of Fig. 6 that the colour of the nodes in the network diagram corresponds to the experimentally-measured resetting kinetics.

To address the concern regarding inconsistent predictions between different model sets, we examined all predictions that flipped (from correct to incorrect and vice versa) from the Dunn et al. 2014 cABN, to the 0.782 cABN and the 0.717 cABN. We only considered predictions in this analysis, and not constraints. We have added a new summary for these results to Table S3 (reproduced below). Overall, the results show that a greater number of predictions flip from incorrect to correct than vice versa, and moreover, once an incorrect prediction has been corrected, it does not flip back as we further refine the models. This supports the phases of model refinement that we carry out.

- **2014 to 0.782:** 8 predictions changed from correct to incorrect. Subsequently, 7 of these 8 cases flipped from incorrect to correct as we refined to the 0.717 cABN. Therefore, we recovered the majority of these ‘lost’ predictions in the 0.717 cABN.
- **2014 to 0.782:** 11 predictions flipped from incorrect to correct. All of these remained correct in the 0.717 cABN.
- **0.782 to 0.717:** 3 predictions flipped from correct to incorrect, while 10 flipped from incorrect to correct.

SUMMARY: FLIPPED PREDICTIONS				
	Correct to incorrect	Flipped to correct in 0.717 model	Incorrect to correct	Flipped to incorrect in 0.717 model
2014 to 0.782	8	7	11	0
0.782 to 0.717	3		10	

4. Finally reviewer 3 makes a very valid point about the interesting possibilities of asynchronous updating that is dismissed by the authors on unrigorous grounds and should definitely be explored.

To address this point, we have incorporated additional analysis to the paper, which is documented in Fig. S8 and in Supplementary Information. As already mentioned, our approach considers a large set of models consistent with a set of imposed constraints, and unrolls deterministic trajectories for each model under a set of experimental conditions. In contrast, other approaches identify a single model and use it to investigate multiple distinct trajectories under the same conditions via simulations based on asynchronous update schemes. To directly compare the two strategies, we used the minimal network from the 0.717 cABN and ran sets of 10,000 simulations (Fig. S8a) under an asynchronous update scheme using BooleanNet, the freely available Boolean network simulator that is also used by Yachie-Kinoshita et al. (referenced by Reviewer 3). We document the results of each set of simulations run under single and double factor forced expressions, single factor knockdowns and gene activation kinetics during EpiSC resetting (Fig. S8b-e). By directly comparing the results from these four different aspects of cellular behaviours, we have sought to provide a comprehensive and unbiased comparison between running a single network under asynchronous updates compared with analysing a large set of possible networks under synchronous updates.

Overall, the minimal model under asynchronous update scheme reached a predictive accuracy of over 63% for each prediction set (Fig. S8f). This level of predictive accuracy lends support to the analysis presented in the main figures, highlighting that our results are not dependent on the assumption of synchronous updates. The results suggest that the 0.717 cABN captures models with behaviours that are relevant to the real biological processes, with individual models possessing high predictive accuracy. Overall, these results reveal that our approach could be exploited to identify candidate network models that can subsequently be used for simulation-based investigation.

2nd Editorial Decision

27th Aug 2018

Thank you for the submission of your revised manuscript (EMBOJ-2018-100003R) to The EMBO Journal. We have carefully assessed your amended study and the point-by-point response provided to the arbitrating advisor's concerns. We have in addition asked this expert to reassess your adjusted manuscript.

Based on the additional comments of the arbitrating expert, together with our reasoning here in the editorial team, we concluded that most of the concerns have been adequately addressed and concur that the level of technical robustness provided is now sufficient for consideration at The EMBO Journal.

We are thus pleased to inform you that your manuscript has been accepted in principle for publication in The EMBO Journal, pending minor revision addressing the remaining data representation issues raised by the advisor.

Please also see below for some changes of the formatting of the manuscript and additional information required as outlined below, which need to be adjusted at re-submission.

ADVISOR'S COMMENTS:

The authors have satisfactorily answered most of my concerns but a major concern and a minor one remain.

1. Regarding the "no prediction" category, it is misleading to just omit that category while reporting the predictive power of the models. For example in Figure S1, model 0.832 is claimed to be 100% accurate but it cannot make any prediction in 2 cases out of 12. More extreme, in table S3, the 2014 cABN network can make predictions in a very limited number of cases (only 11% for the LIF+CHIR condition).

If the authors want to keep reporting the figures excluding the "no prediction" category, they have to explicitly state: no prediction could be made in XX% of cases (in the text but also in the figures, supplementary figures and tables). The preferred and more valid option would be to revise the reported figures in order to include "no prediction" as incorrect.

In general, the presence of a "no prediction" category and the relative prevalence of it depending on the model is not sufficiently discussed in the text.

2. Using the same layout but the networks in Supplementary Figures 1, 2 and 4 have to be updated for consistency.

2nd Revision - authors' response

25th Sep 2018

We thank the editor and reviewer's further assessment and consideration. We are very pleased to hear that our manuscript is accepted in principle for The EMBO Journal.

Point by point response:

The authors have satisfactorily answered most of my concerns but a major concern and a minor one remain.

1. Regarding the "no prediction" category, it is misleading to just omit that category while reporting the predictive power of the models. For example in Figure S1, model 0.832 is claimed to be 100% accurate but it cannot make any prediction in 2 cases out of 12. More extreme, in table S3, the 2014 cABN network can make predictions in a very limited number of cases (only 11% for the LIF+CHIR condition). If the authors want to keep reporting the figures excluding the "no prediction" category, they have to explicitly state: no prediction could be made in XX% of cases (in the text but also in the figures, supplementary figures and tables). The preferred and more valid option would be to revise the reported figures in order to include "no prediction" as incorrect.

We appreciate the concern from the reviewer regarding the "no prediction" category. We now provide the percentage of "no prediction" for each condition tested in Table S3. In addition, we calculated the overall percentage of "no prediction" for all our models, which is shown in Fig. 7f and Table S3.

We should also stress that in all panels presenting our predictions (Fig. 1g, 2a, 6a, 7a-b) we clearly indicated when predictions could not be made (indicated as "No prediction" or "Some models").

Concerning the alternative presentation of "no prediction" as incorrect, we respectfully disagree. When we formulate predictions we test whether, for instance, gene X will be active upon perturbation Y. More specifically, we test this for each of the concrete networks in the cABN. Accordingly, the answer can be:

- a- gene X will be definitely Active – it is active in all networks
- b- gene X will be definitely Inactive – it is inactive in all networks
- c- gene X can be either Active or Inactive (i.e. no prediction) – it is active in some but not all networks.

We then perform experimental validation and only cases "a" and "b" can be proved or disproved, while the answer "c" cannot be wrong, because the model is consistent with both scenarios (gene X active or inactive). In this case experimental data are used to further constrain the model (if gene X was found to be active we will then impose it as a constraint).

The percentage of predictions made does not concern the accuracy of a model, but rather its predictive power. As correctly pointed out there is one experiment of the 2014 cABN where predictions are made only for 11% of cases. When we look globally at all experiments for this cABN we observe that predictions were not made in 63.38% of cases, with the remaining predictions yielding an accuracy of 66.67%. Our final 0.717 cABN had 27.05% "no predictions", with the remaining predictions yielding an accuracy of 77.4%. Showing and discussing the two parameters separately is critical to appreciate the improvement observed during refinement of the cABN. We are grateful to the reviewer for highlighting this issue.

In general, the presence of a "no prediction" category and the relative prevalence of it depending on the model is not sufficiently discussed in the text.

We have included additional discussion regarding the "no prediction" category in the Discussion section, including its relative prevalence, how it arises and its distinction from the "incorrect" category.

2. Using the same layout but the networks in Supplementary Figures 1, 2 and 4 have to be updated for consistency.

We have updated these figures accordingly.

3rd Editorial Decision

2nd Oct 2018

Thank you for submitting your revised manuscript for consideration by The EMBO Journal. I have assessed your changes and find that most of the minor concerns left have been sufficiently addressed.

We however still need you to address few minor issues regarding formatting and data-methods annotation and documentation with the study, as outlined below.
Once we receive the updated files, we will move on swiftly with formal acceptance of your study.

Corresponding Author Name: Graziano Martello

Journal Submitted to: The EMBO Journal

Manuscript Number: EMBOJ-2018-100003R